# CALM: Co-evolution of Algorithms and Language Model for Automatic Heuristic Design

**Ziyao Huang[1,2], Weiwei Wu[3], Kui Wu[4], Jianping Wang[1,2,*], Wei-Bin Lee[5]**

[1]City University of Hong Kong
[2]City University of Hong Kong Matter Science Research Institute (Futian)
[3]Southeast University [4]University of Victoria [5]Hon Hai Research Institute

`zhuang88@cityu.edu.hk, weiweiwu@seu.edu.cn, wkui@uvic.ca`
`jianwang@cityu.edu.hk, wei-bin.lee@foxconn.com`

## Abstract

Tackling complex optimization problems often relies on expert-designed heuristics, typically crafted through extensive trial and error. Recent advances demonstrate that large language models (LLMs), when integrated into well-designed evolutionary search frameworks, can autonomously discover high-performing heuristics at a fraction of the traditional cost. However, existing approaches predominantly rely on verbal guidance, i.e., manipulating the prompt generation process, to steer the evolution of heuristics, without adapting the underlying LLM. We propose a hybrid framework that combines verbal and numerical guidance, the latter achieved by fine-tuning the LLM via reinforcement learning (RL) based on the quality of generated heuristics. This joint optimization allows the LLM to co-evolve with the search process. Our method outperforms state-of-the-art (SOTA) baselines across various optimization tasks, running locally on a single 24GB GPU using a 7B model with INT4 quantization. It surpasses methods that rely solely on verbal guidance, even when those use significantly more powerful API-based models. The code is available at: `https://github.com/whxru/CALM`.

## 1 Introduction

Complex optimization problems are prevalent in real-world applications, including logistics (Duan et al., 2022; Tresca et al., 2022), scheduling (Mihoubi et al., 2021; Palacio et al., 2022), and transportation (Dahmani et al., 2024; Pereira et al., 2021). Traditionally, solving these problems relies heavily on manually crafting high-quality heuristics, a labor-intensive process requiring substantial expert knowledge. Given the limitations of this manual approach, Automatic Heuristic Design (AHD) emerged to streamline heuristic generation. Nevertheless, classic AHD approaches like Genetic Programming (GP) (Burke et al., 2009) still depend significantly on human-defined problem-specific components, limiting the search space and flexibility.

Recently, the advent of Large Language Models (LLMs) has introduced promising avenues for AHD by employing LLMs as heuristic generators and evolutionary computing (EC) techniques as a search framework. In this paradigm, heuristics generated by LLMs are iteratively evaluated through a predefined simulation framework, and superior heuristics inform subsequent generation prompts, thus creating a feedback-driven evolutionary loop (Liu et al., 2024a). Nevertheless, existing LLM-based AHD methods predominantly keep the underlying LLM untouched and merely guide heuristic evolution via textual prompt manipulations, referred to as "verbal gradients" (Ye et al., 2024). Consequently, these methods inherently neglect the opportunity of tuning and enhancing the generative capability of LLM based on the feedback from heuristic designs.

We propose Co-evolution of Algorithms and the Language Model (CALM) to capture this opportunity. CALM drastically differs from the state-of-the-art (SOTA) (Liu et al., 2024a; Ye et al., 2024; Dat et al., 2025; Zheng et al., 2025) by enabling the LLM to co-evolve alongside heuristic designs. This co-evolution is made possible by treating the heuristic generation process not only as a target of

---

*Corresponding author.

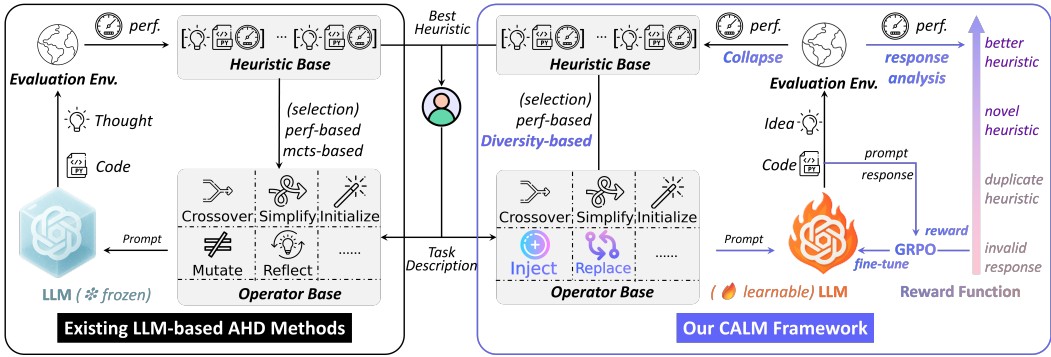

Figure 1: Pipeline of existing LLM-based AHD methods (Romera-Paredes et al., 2024; Ye et al., 2024; Dat et al., 2025; Zheng et al., 2025) under a fixed LLM and our new approach CALM that enables the co-evolution of LLM in the iterative heuristic search process. New components are presented in bright colors.

optimization but also as a rich source of training data. As heuristics are continually proposed, evaluated, and selected based on their performance, the evolutionary loop naturally produces abundant prompt-response-performance triplets. These data points are highly informative, as each heuristic's effectiveness provides an implicit signal about the utility of the underlying generation process. By using this signal as feedback for reinforcement learning (RL), we can fine-tune the LLM, thereby applying what we term "numerical gradients" to adapt the model itself. This co-evolution approach unlocks a new dimension of adaptability, allowing the LLM to internalize characteristics of successful heuristics and improve its future generations.

CALM is one of the first LLM-based AHD frameworks that jointly optimize both the prompt generation process and the LLM model itself, overcoming the limitations of fixed-model approaches. For prompt generation, CALM introduces a suite of evolutionary operators, including fine-granularity mutation operators (injection and replacement) and a diversity-aware crossover operator, that promote meaningful and diverse heuristic variations while preserving structural coherence. Furthermore, a simple yet effective collapse mechanism is developed to help escape the local optima. For model improvement, CALM employs a memory-efficient RL algorithm GRPO (Shao et al., 2024) with a carefully designed reward function to enable efficient fine-tuning. Experimental results demonstrate that our new approach can discover heuristics that beat existing SOTA baselines (Liu et al., 2024a; Ye et al., 2024; Zheng et al., 2025), while running entirely on a local computer with a single 24GB GPU, in contrast to prior methods that depend heavily on commercial LLM APIs.

## 2 RELATED WORK

As our approach centers on fine-tuning LLMs by RL for solving optimization problems, we review relevant literature in both RL and LLMs applied to optimization. Additional related topics, including LLMs for code generation and RL-based LLM fine-tuning, are discussed in Appendix B.

**RL for Optimization Problems.** Existing RL-based methods for optimization can be broadly categorized by the role the learned policy plays: *(1) Instance-Level Solution Generator.* Deep RL has been widely adopted to learn policies for solving specific optimization instances (Kwon et al., 2020; Pan et al., 2023; Bi et al., 2024). However, these methods differ fundamentally from LLM-based AHD methods, as they directly produce solutions rather than design the algorithms that generate them. The LLM-based AHD approach operates at *a meta level*, seeking to learn the algorithmic structure that produces solutions. This distinction also applies to the broader class of Neural Combinatorial Optimization (NCO) (Luo et al., 2024; Xiao et al., 2024; Sui et al., 2024; Zheng et al., 2023), where models are trained to directly solve instances. Moreover, NCO methods often require explicit adaptation to handle problem scales not seen during training, whereas our method generalizes more naturally to new scales. *(2) Heuristic Generator.* Some RL-based methods target meta-level search to discover heuristics instead of instance-level solutions. For example, AlphaDev (Mankowitz et al., 2023) learns to combine low-level operations to discover faster sorting algorithms, and Yi et al.

(2022) searches for high-performing metaheuristics from predefined algorithmic components. While having similar goals, these approaches rely heavily on hand-engineered building blocks, akin to traditional AHD frameworks (Pillay and Qu, 2018; Sánchez-Díaz et al., 2021; Burke et al., 2009). In contrast, LLM-based method reduces manual intervention by leveraging LLMs to explore an open-ended heuristic space with minimal prior specification.

**LLM for Optimization Problems.** Studies in this area fall into two categories depending on how LLMs are employed: *(1) Instance-Level Solution Generator.* Several works (Abgaryan et al., 2024; Jiang et al., 2024; Wu et al., 2024) prompt LLMs with instance-specific inputs for direct solution generation. LLM-based methods in this category focus on discovering reusable heuristics. Moreover, methods such as that proposed by Jiang et al. (2024) and Wu et al. (2024) keep LLM parameters frozen, and Abgaryan et al. (2024) fine-tune the model using supervised labels from an existing solver (Perron and Furnon, 2024). In contrast, our approach requires no imitation dataset, enabling its application to problems lacking established solvers. *(2) Heuristic Generator.* LLM-based AHD methods (Liu et al., 2023a; Chen et al., 2025; Romera-Paredes et al., 2024; Liu et al., 2024a; Ye et al., 2024; Liu et al., 2024b; Dat et al., 2025; Zheng et al., 2025; Novikov et al., 2025) repeatedly ingest information about the current elite heuristicstypically their natural-language descriptions, source code, and performance scoresand, via fixed prompt templates that mimic genetic operators, produce new candidate heuristics. Those candidates are then executed and evaluated, and the resulting feedback is fed back into the prompt, forming an evaluategenerate loop that continues until the evaluation budget is exhausted. Additionally, some recent studies have also explored reduction techniques (Thach et al., 2025), trajectory-based analysis (Yang et al., 2025), multi-objective optimization (Yao et al., 2025), to further enhance AHD. Wu et al. (2025) have examined how to abstract core components from elite heuristics and combine them with LLM-based fitness prediction for AHD. However, prior work keeps the LLM static. Our approach improves this by continuously fine-tuning the LLM using prompt-response-performance tuples from the evolutionary process, enhancing future heuristic generation. Notably, there are concurrent explorations on fine-tuning LLMs for AHD (Surina et al., 2025; Liu et al., 2025). These studies provide valuable insights into how preference-based fine-tuning methods such as DPO (Rafailov et al., 2023) can improve heuristic discovery. Our work adopts a different approach by employing the score-based RL algorithm (Shao et al., 2024) to fine-tune LLMs for AHD, and further introduces specialized designs such as fine-granularity operators to enhance the fine-tuning process through prompt manipulation.

## 3 PRELIMINARY

### 3.1 LLM-BASED AHD

Let $P$ be a problem with input space $\mathcal{I}$ and solution space $\mathcal{S}$, and let a *heuristic* be a function $h : \mathcal{I} \rightarrow \mathcal{S}$. Given a training set $D \subset \mathcal{I}$ and an objective $f : \mathcal{S} \rightarrow \mathbb{R}$ (lower is better), the performance of a heuristic is $g(h) = \mathbb{E}_{x \in D}[-f(h(x))]$. Let $\mathcal{H}$ denote the space of all feasible heuristics. The objective of AHD is to identify the optimal heuristic within this space, i.e., $h^* = \arg\max_{h \in \mathcal{H}} g(h)$.

LLM-based AHD is AHD where LLM serves as a heuristic generator. In practice, the LLM is charged with designing the core decision function of a solver. For example, on tasks like the Traveling Salesman Problem (TSP) or the Capacitated Vehicle Routing Problem (CVRP), an LLM-based AHD method might generate a function, which selects the next city to visit or constructs an edge-desirability matrix to guide solution search within an Ant Colony Optimization (ACO) framework.

### 3.2 GRPO

GRPO (Shao et al., 2024) is a recent RL algorithm that has proven effective in training LLMs, as evidenced by its application in models such as DeepSeek-R1. GRPO starts from an initial model $\pi_\theta$ and a reward function denoted by $r_\phi(q, o)$ that maps the prompt $q$ and the generated response $o$ to a scalar. At the beginning of each training round, it snapshots $\pi_\theta$ as a reference model $\pi_{\text{ref}}$. Then, it split all task prompts into multiple batches. When training for each prompt batch $\mathcal{D}_b$, it first snapshots $\pi_\theta$ as $\pi_{\text{old}}$. For each task prompt $q \in \mathcal{D}_b$, it samples a group of $G$ responses $\{o_i\}_{i=1}^G \sim \pi_{\theta_{\text{old}}}$ and computes rewards $\boldsymbol{r} = \{r_i = r_\phi(q, o_i)\}_{i=1}^G$ for each prompt-response pair. Subsequently, it computes the advantage $\hat{A}_{i,t}$ for each token $t$ in response $i$ as the normalized reward

$(r_i - \text{mean}(\boldsymbol{r}))/\text{std}(\boldsymbol{r})$. The model parameters $\theta$ are updated by maximizing the following objective function:

$$\mathcal{J}_{\text{GRPO}}(\theta) = \mathbb{E}_{\left[q \sim \mathcal{Q}, \{o_i\} \sim \pi_{\theta_{\text{old}}}\right]}$$

$$\frac{1}{G}\sum_{i=1}^{G}\frac{1}{|o_i|}\sum_{t=1}^{|o_i|}\left\{\min\left[\hat{r}_{i,t}\hat{A}_{i,t},\ \text{clip}\left(\hat{r}_{i,t}, 1-\varepsilon, 1+\varepsilon\right)\hat{A}_{i,t}\right] - \beta\,\mathbb{D}_{\text{KL}}\left[\pi_\theta\|\pi_{\text{ref}}\right]\right\}, \quad (1)$$

where $\epsilon$ and $\beta$ are hyper-parameters, $\hat{r}_{i,t} = \pi_\theta(o_{i,t} \mid q, o_{i,<t})/\pi_\theta^{\text{old}}(o_{i,t} \mid q, o_{i,<t})$, and the KL divergence term is computed using an unbiased estimator (Schulman, 2020) with guaranteed positivity. GRPO uses the group mean reward as a baseline to eliminate the need for an auxiliary value network, thereby reducing memory requirements. Additionally, the clipping mechanism combined with KL divergence regularization ensures stable and conservative updates.

## 4 METHODOLOGY

To explore the benefit of RL-based fine-tuning for discovering higher-quality heuristics in LLM-based AHD, we introduce CALM, a novel framework that integrates both verbal and numerical guidance in evolutionary heuristic search. As shown in Fig. 1, CALM maintains a pool of heuristics, each with its own idea, code, and performance. At every round, CALM draws a feasible evolutionary operator to produce a new prompt $q$. Subsequently, $G$ responses are sampled from the local LLM $\pi_\theta$, which are then evaluated. Based on the evaluation results, rewards are assigned to each response for GRPO to train the LLM, and new feasible heuristics are added to the pool. Consequently, CALM returns the best-so-far heuristic after running $T$ rounds. Next, we elaborate on the critical techniques in CALM: prompt generation, collapse mechanism, and the reward function.

### 4.1 PROMPT GENERATION

CALM provides several evolutionary operators: injection, replacement, crossover, simplification, and initialization. Prompts are predominantly generated by the selected operator and heuristics sampled from maintained pools. The initialization operator is an exception, as it does not require heuristics from the pool. Next, we elaborate on the heuristic sampling method and operators[1].

**Heuristic Sampling Method.** The heuristic sampling approach varies for the crossover operator, details of which will be provided when introducing this operator. For the remaining operators, i.e., injection, replacement, and simplification, the heuristics are selected based on their performance rankings like (Liu et al., 2024a). Specifically, the probability of sampling a heuristic $h$ is inversely proportional to its rank in the current pool (i.e., proportional to $1/\text{rank}_p(h)$). Heuristics ranked below a threshold, defined as the population size, are assigned a probability of zero.

**Fine-Granularity Mutation Operators: Injection & Replacement.** GRPO assigns an advantage score to each token based on the relative reward of the full response compared to others from the same prompt. This means each part of a heuristic is encouraged or penalized depending on the quality of the whole. However, heuristic performance can shift dramatically with changes to even a single sub-component, making uniform treatment of all parts in terms of gradient direction unreliable.

While cumulative gradient updates can correct misattributed rewards or penalties for the same token appearing in different responses, we aim to further boost this process. To this end, we introduce two novel operators that enable more precise control over heuristic variations. These operators encourage the LLM to retain more common parts while introducing meaningful modifications to the input heuristic (See Appendix E for examples). Consequently, GRPO is expected to more effectively identify the contribution of individual structural changes. The two newly designed operators are:

*Injection.* Given an existing heuristic, the injection operator prompts the LLM to incorporate a new component into it. Additionally, a concise description of the new component must be included in the response. All component descriptions are stored, and subsequent applications of the injection operator require the LLM to introduce components distinct from those previously saved, promoting diversity in generated heuristics. Compared to prior LLM-based AHD methods (Zheng et al., 2025;

---

[1]The complete algorithm and prompt details are in Appendix C and D, respectively.

Liu et al., 2024a): (1) To explore under-explored heuristic designs, CALM provides the code of only one base heuristic together with compact summaries of components that have already been explored, whereas prior methods typically supply the full code of multiple existing heuristics to prompt the LLM to produce a different one. This design allows more references to be accommodated within the LLM's context window. (2) Saved component descriptions are globally accessible and not limited to the currently sampled heuristics; (3) Prior methods often require entirely new heuristics, while our approach focuses on more granular modifications; (4) When the number of heuristics is below the population size, the sampling probability of the injection operator is increased to encourage exploration in the phase of population expansion.

***Replacement.*** Given an existing heuristic, the replacement operator prompts the LLM to rewrite an existing component under a specific instruction. There are three distinct instructions, and each time the replacement operator is applied, one is randomly sampled for the given heuristic. While the "rewrite hyper-parameter" instruction is also present in prior studies (Liu et al., 2024a; Zheng et al., 2025), CALM introduces two novel instructions: (1) Rewrite an instance-independent decision rule as an instance-dependent oneto improve the heuristics adaptability to varying problem contexts; (2) Rewrite a fragment that assigns equal or near-equal credit to all candidates as one that differentiates credit based on contextual performanceto encourage more effective prioritization and refined decision-making.

**DiversityAware Crossover.** To balance exploitation and exploration, each crossover invocation randomly chooses between (1) *performancebased*: sample both parents by performance rank; and (2) *diversitybased*: sample the first parent $h_{c,1}$ by performance rank and the second from all retained heuristics with probability inversely proportional to diversity rank (larger diversity is better). Specifically, let $\text{idea\_token}(\cdot)$ denote the set of unique tokens in a heuristic's idea, the diversity is: $\text{div}(h_{c,1}, h) = |\text{idea\_token}(h) \setminus \text{idea\_token}(h_{c,1})|/|\text{idea\_token}(h)|$. This hybrid mechanism ensures that at least one parent heuristic is of high quality, while the second parent is either high-performing or structurally novel. The diversity-aware selection expands the evolutionary search space and leverages underutilized heuristics, potentially unlocking novel strategies that might otherwise be overlooked due to suboptimal early performance. More discussions are moved to Appendix F.

**Simplification Operator.** As heuristic structures grow increasingly complex through repeated applications of injection, crossover, and replacement, there is a risk of accumulating redundant or unnecessarily verbose components. The simplification operator counterbalances this tendency by prompting the LLM to produce a more concise and effective version of a given heuristic.

**Initialization Operator.** In cases where there is no heuristic in the pool (e.g., no initial/seeding function is provided), this operator is invoked to prompt the LLM to generate new heuristics.

## 4.2 COLLAPSE MECHANISM

**Why to Collapse.** A key reason LLM-based evolutionary heuristic search can succeed is that prompts containing better-performing heuristics tend to guide the LLM toward generating even stronger ones. This creates a self-reinforcing feedback loop, gradually evolving a population of increasingly effective heuristics. However, this process can also lead to inbreeding and premature convergence: over time, the population becomes dominated by minor variations of the current best-performing heuristic. When this state persists without meaningful breakthroughs, the search risks becoming trapped in a local optimum, a classic challenge in evolutionary computing (Eshelman, 1991).

**How to Collapse.** As a remedy, CALM introduces a proactive collapse mechanism that resets the search process when it detects stagnation, allowing the system to escape local optima and reinitiate meaningful exploration. Specifically, when the search has plateauedcharacterized by a prolonged lack of performance improvementwe reset the population by discarding all heuristics except two: the original seed algorithm and the current best-performing heuristic. These two retained heuristics jointly serve as the seed algorithms for the new search process, grounding it in past progress while freeing it from the genetic redundancy accumulated in the previous population.

**When to Collapse.** Once the heuristic pool reaches its target population size, CALM begins tracking stagnation using a no-breakthrough counter $c_n$, initialized to zero. This counter records the number

of consecutive prompt roundseach involving $G$ sampled responsesthat fail to yield a globally superior heuristic. If any sampled heuristic in a round surpasses all previous ones in performance, $c_n$ is reset to zero; otherwise, it increments by one.

To escape local optima, CALM introduces a probabilistic collapse mechanism based on this counter. At the end of each round, collapse is triggered if: $\mathrm{random}(0,1) < c_n \delta_0$ or $c_n \geq C$, where $\delta_0 \ll 1$ controls the rate at which collapse probability grows, and $C$ is a hard cap ensuring collapse happens by the $C$-th stagnation step at the latest. To aid in hyperparameter selection, we further provide an analytical approximation for the expected number of rounds before collapse is triggered:

$$\mathbb{E}\left[c_n \mid \text{collapse}, C > \frac{1}{\delta_0}\right] \approx \sqrt{\frac{\pi}{2\delta_0}}. \tag{2}$$

This collision of a risingprobability rule with a fixed maximum fosters a balance between giving the search plenty of room to improve and ensuring it doesnt stall infinitely. A detailed proof and discussion about the benefit of the mechanism can be found in Appendix G.

### 4.3 DESIGN OF REWARD FUNCTION

The reward function assigns a score to each LLM-generated response, enabling the RL algorithm to update the LLMs parameters and progressively improve its outputs. In AHD, we aim for responses that yield feasible, novel, and high-performing heuristics. To guide this process, we adopt a *progressive* scoring scheme that assigns increasing scores across the following categories: (1) infeasible responses that fail to produce valid heuristics, (2) duplicate heuristics offering no new insights, (3) new heuristics, and (4) new high-performing heuristics.

For each invalid response, we assign a reward bounded below by a scalar $r_{\text{invalid}} \in (-1, 0)$. Rewards for valid heuristics are defined relative to this bound, ensuring that valid outputs always score higher.

For valid heuristics, performance serves as the primary learning signal. However, because the quality of the generated heuristic is influenced by the promptparticularly its base heuristicswe avoid attributing full credit or blame to the LLM alone. Instead, we reward improvements relative to the best base heuristic in the prompt, ensuring that learning reflects meaningful gains rather than prompt bias. Specifically, let $H$ denote the set of base heuristics used to construct prompt $q$, and $h_{\text{new}}$ be the heuristic parsed from the LLMs output $o$. We define the top base heuristic as $h_{\text{t\_base}} = \arg\max_{h \in H} g(h)$, and measure relative performance via:

$$\Delta(h_{\text{new}}, h_{\text{t\_base}}) = \mathrm{clip}\left(\frac{|g(h_{\text{new}}) - g(h_{\text{t\_base}})|}{\min\{|g(h_{\text{new}})|, |g(h_{\text{t\_base}})|\}}, 0, 1\right). \tag{3}$$

Let $\alpha_1, \alpha_2 \in (0, 1)$ and $\alpha_1 > \alpha_2$, the reward function $r_\phi(q, o \mid h_{\text{new}}, h_{\text{t\_base}})$ is then defined as:

$$r_\phi(q, o \mid h_{\text{new}}, h_{\text{t\_base}}) = \begin{cases} \alpha_1 r_{\text{invalid}}, & \text{if } \exists h \in H \text{ s.t. } g(h) = g(h_{\text{new}}); \\ \alpha_2 r_{\text{invalid}} \cdot \Delta(h_{\text{new}}, h_{\text{t\_base}}), & \text{if } g(h_{\text{new}}) < g(h_{\text{t\_base}}); \\ 1 + \Delta(h_{\text{new}}, h_{\text{t\_base}}), & \text{if } g(h_{\text{new}}) > g(h_{\text{t\_base}}). \end{cases} \tag{4}$$

Under the reward function above, the reward is primarily determined by whether the new heuristic improves over the best base heuristic or not, with the relative performance gap further modulating the strength of the reward or penalty. When the generated heuristic is identical in performance to an existing base heuristic, a small but consistent reward ($\alpha_1 r_{\text{invalid}}$) is given to discourage trivial reproduction. If the new heuristic underperforms relative to the best base, a scaled negative reward is applied, while genuine improvements yield strictly positive rewards starting from 1.

## 5 EXPERIMENTS

**Implementation Details of CALM.** We build CALM on Unsloth (Daniel Han and team, 2023) and employ an INT4-quantized Qwen2.5-7B-Instruct model (Yang et al., 2024), fine-tuning just 1.15% of its weights. INT4 compression cuts memory usage up to 8Œ versus FP32 but degrades precision. According to Yang et al. (2024), performance ranks as follows: GPT-4o-mini $\approx$ Qwen2.5-Turbo > Qwen2.5-14B-Instruct > Qwen2.5-7B-Instruct > Qwen2.5-7B-Instruct-INT4. The 14B and 7B

Instruct models share the same architecture, so the larger parameter count drives the 14Bs edge over the 7B, while quantization further reduces the 7Bs accuracy. Consequently, GPT-4o-minibased baselines retain a clear advantage in raw accuracy over our lean, resource-efficient setup. More implementation details can be found in Appendix H.

**Optimization Tasks.** Existing LLM-based methods can demonstrate near-optimal or optimal performance on some benchmark problems, such as TSP (Liu et al., 2024a; Ye et al., 2024; Zheng et al., 2025) (aided by ACO solvers) and knapsack problem (KP) (Zheng et al., 2025), leaving little room for further improvement. Therefore, we focus on tasks that remain challenging for LLM-based AHD as follows: Online Bin Packing (OBP) problem and TSP under step-by-step construction task, CVRP and Orienteering Problem (OP) under an ACO search framework. Detailed problem descriptions can be found in Appendix H.3.

**Baselines.** To evaluate CALM, we compare its designed heuristics against the following baselines: (1) hand-crafted heuristics such as Best-Fit (Kenyon, 1995) for OBP, Greedy-Construct (GC) (Rosenkrantz et al., 1977) for TSP, and ACO (Blum, 2005) for CVRP and OP; (2) Nerual Combinatorial Optimization (NCO) methods including POMO (Kwon et al., 2020) and DeepACO (Ye et al., 2023); and (3) LLM-based AHD approaches like FunSearch (Romera-Paredes et al., 2024), EoH (Liu et al., 2024a), ReEvo (Ye et al., 2024), HSEvo (Dat et al., 2025), OpenEvolve (Sharma, 2025), MCTS-AHD (Zheng et al., 2025), and EvoTune (Surina et al., 2025). Notably, AlphaEvolve (Novikov et al., 2025) does not release its official source code. OpenEvolve, developed by an independent group, is one of the most popular open-source reimplementations of AlphaEvolve. To ensure a fair comparison, we align CALM and all LLM-based AHD baselines with consistent settings, including shared seed heuristics that are directly adopted from Zheng et al. (2025), identical training datasets for evaluating heuristic performance, and comparable evaluation budgets–specifically, 1,000 heuristic evaluations for baselines and a fixed budget of 2,000 LLM queries for CALM across all tasks except OBP. Notably, prior AHD methods typically conduct 2,000 heuristic evaluations using over 4,000 queries for OBP, whereas CALM operates under a fixed budget of 2,000 queries.

## 5.1 OVERALL RESULTS

**OBP.** We train and evaluate CALM on the same dataset used by Zheng et al. (2025), which includes four training instances at varying scales and five testing instances spanning six different scalestwo of which are out-of-domain and not seen during training. Results in Table 1 show that CALM consistently outperforms all baseline methods in terms of average optimality gap across the full test set. It can achieve superior performance on out-of-domain and in-domain scales. Remarkably, CALM achieves a zero gap in set 1k_500, indicating exact optimal solutions at that scale.

Table 1: Average optimality gaps of heuristics for OBP over three runs. All methods are trained and evaluated on the same datasets as Zheng et al. (2025), with gaps measured relative to the lower bound by Martello and Toth (1990). Test sets whose scale matches the training distribution are underlined. Format: 1k_100 denotes instances with $1,000$ items and a bin capacity of 100.

| | | | **Online Bin Packing (OBP)** | | | | |
|---|---|---|---|---|---|---|---|
| Test sets | 1k_100 | 1k_500 | 5k_100 | 5k_500 | 10k_100 | 10k_500 | Avg. |
| Best Fit | 4.77% | 0.25% | 4.31% | 0.55% | 4.05% | 0.47% | 2.40% |
| First Fit | 5.02% | 0.25% | 4.65% | 0.55% | 4.36% | 0.50% | 2.56% |
| | | *LLMbased AHD:* **GPT-4o-mini** (w/o. GRPO) | | | | | |
| FunSearch | **2.45%** | 0.66% | 1.30% | 0.25% | 1.05% | 0.21% | 0.99% |
| EoH | 2.69% | 0.25% | 1.63% | 0.53% | 1.47% | 0.45% | 1.17% |
| ReEvo | 3.94% | 0.50% | 2.72% | 0.40% | 2.39% | 0.31% | 1.71% |
| HSEvo | 2.64% | 1.07% | 1.43% | 0.32% | 1.13% | 0.21% | 1.13% |
| OpenEvolve | 4.84% | 0.25% | 4.28% | 0.55% | 4.07% | 0.47% | 2.41% |
| MCTS-AHD | **2.45%** | 0.50% | 1.06% | 0.32% | 0.74% | 0.26% | 0.89% |
| CALM (Ours) | 2.78% | 0.29% | **0.83%** | 0.28% | **0.50%** | 0.24% | 0.82% |
| | | *LLMbased AHD:* **Qwen2.5-7B-Instruct-INT4** (w/. GRPO) | | | | | |
| EvoTune | 4.67% | 0.25% | 4.23% | 0.55% | 4.11% | 0.60% | 2.40% |
| CALM (Ours) | 2.55% | **0.00%** | 0.85% | **0.17%** | 0.56% | **0.14%** | **0.71%** |

**TSP.** CALM is trained on the same dataset used by Zheng et al. (2025): a training set of 64 TSP instances with $N = 50$ nodes and three test sets of 1,000 instances each at $N = 50, 100$, and 200. As shown in Table 2, CALM-constructed heuristics outperform all LLM-based baselines on both out-of-domain test sets and achieve the second-best LLM-based result on the in-domain set. Notably, at the largest scale, CALM surpasses the NCO baseline POMO, which requires per-scale training.

Table 2: Performance on TSP, averaged over three runs. Methods are evaluated on three test sets of 1,000 instances each, using the same training and testing datasets as by Zheng et al. (2025). In-domain scales (i.i.d. to training) are underlined. Optimal tours are from LKH (Lin and Kernighan, 1973). Best LLM-based results are shaded, overall best in bold.

| | **Traveling Salesman Problem (TSP)** | | | | | |
| | N=50 | | N=100 | | N=200 | |
| Methods | Obj.↓ | Gap↓ | Obj.↓ | Gap↓ | Obj.↓ | Gap↓ |
| Optimal | 5.675 | – | 7.768 | – | 10.659 | – |
| GC | 6.959 | 22.62% | 9.706 | 24.94% | 13.461 | 26.29% |
| POMO | **5.697** | **0.39%** | **8.001** | **3.01%** | 12.897 | 20.45% |
| *LLM-based AHD:* **GPT-3.5-turbo (w/o. GRPO)** | | | | | | |
| FunSearch | 6.683 | 17.75% | 9.240 | 18.95% | 12.808 | 19.61% |
| EoH | 6.390 | 12.59% | 8.930 | 14.96% | 12.538 | 17.63% |
| MCTS-AHD | 6.346 | 11.82% | 8.861 | 14.08% | 12.418 | 16.51% |
| *LLM-based AHD:* **GPT-4o-mini (w/o. GRPO)** | | | | | | |
| FunSearch | 6.357 | 12.00% | 8.850 | 13.93% | 12.372 | 15.54% |
| EoH | 6.394 | 12.67% | 8.894 | 14.49% | 12.437 | 16.68% |
| OpenEvolve | 6.281 | 10.68% | 8.719 | 12.25% | 12.148 | 13.96% |
| MCTS-AHD | 6.225 | 9.69% | 8.684 | 11.79% | 12.120 | 13.71% |
| CALM (Ours) | 6.273 | 10.54% | 8.691 | 11.88% | 12.104 | 13.56% |
| *LLM-based AHD:* **Qwen2.5-7B-Instruct-INT4 (w. GRPO)** | | | | | | |
| EvoTune | 6.267 | 10.43% | 8.777 | 12.99% | 12.429 | 16.60% |
| CALM (Ours) | 6.244 | 10.04% | 8.668 | 11.58% | **12.088** | **13.41%** |

**CVRP.** CALM is trained on 10 instances as in (Zheng et al., 2025) with $N = 50$ nodes using the ACO framework, and evaluated on three test sets of 64 instances each at $N = 50, 100$, and 200, following the same generation protocol. During both training and testing, the number of ants and iterations is fixed to 30 and 100, respectively. As shown in Table 3, CALM consistently outperforms all LLM-based baselines across all test sets, including both the in-domain and out-of-domain ones.

**OP.** CALM is trained 5 OP instances with $N = 50$ nodes using the ACO framework and evaluated on three test sets of 64 instances each at $N = 50, 100$, and 200, following the generation protocol in HSEvo (Dat et al., 2025). Both training and testing use a fixed configuration of 20 ants and 50 iterations. As reported in Table 3, CALM consistently outperforms all other LLM-based baselines on the out-of-domain scales. As for the in-domain scale, it still outperforms EoH and the most recent approach, MCTS-AHD and EvoTune.

Table 3: Performance of ACO-based heuristics on CVRP and OP, averaged over three runs. All methods are evaluated on three test sets of 64 randomly generated instances each, following the setup in (Zheng et al., 2025) and (Dat et al., 2025), respectively. Optimal solutions are approximated using DeepACO with significantly more ants and iterations than those in the baseline configurations.

| | **CVRP** | | | | | | **OP** | | | | | |
| | N=50 | | N=100 | | N=200 | | N=50 | | N=100 | | N=200 | |
| Methods | Obj.↓ | Gap↓ | Obj.↓ | Gap↓ | Obj.↓ | Gap↓ | Obj.↑ | Gap↓ | Obj.↑ | Gap↓ | Obj.↑ | Gap↓ |
| Optimal | 8.888 | | 14.932 | | 27.159 | | 19.867 | | 36.392 | | 63.380 | |
| ACO | 18.581 | 109.05% | 30.107 | 101.63% | 37.590 | 40.69% | 13.354 | 32.69% | 24.131 | 33.69% | 37.586 | 40.69% |
| *LLM-based AHD:* **GPT-4o-mini (w/o. GRPO)** | | | | | | | | | | | | |
| EoH | 9.894 | 11.32% | 16.953 | 13.54% | 30.314 | 11.62% | 13.388 | 32.61% | 24.154 | 33.63% | 37.319 | 41.12% |
| ReEvo | 9.558 | 7.54% | 16.350 | 9.50% | 29.219 | 7.58% | 15.103 | 23.98% | 30.523 | 16.13% | 53.807 | 15.10% |
| HSEvo | 9.431 | 6.11% | 16.396 | 9.81% | 29.520 | 8.69% | 15.082 | 24.08% | 30.454 | 16.32% | 53.260 | 15.97% |
| OpenEvolve | 10.077 | 13.37% | 17.418 | 16.65% | 31.190 | 14.84% | 14.314 | 27.95% | 28.336 | 22.13% | 48.576 | 23.36% |
| MCTS-AHD | 9.372 | 5.44% | 15.974 | 6.98% | 28.434 | 4.70% | 14.847 | 25.27% | 30.163 | 17.12% | 53.024 | 16.34% |
| CALM (Ours) | 9.404 | 5.81% | 16.046 | 7.46% | 28.713 | 5.72% | 15.017 | 24.41% | 30.294 | 16.76% | 53.098 | 16.22% |
| *LLM-based AHD:* **Qwen2.5-7B-Instruct-INT4 (w/. GRPO)** | | | | | | | | | | | | |
| EvoTune | 9.405 | 5.82% | 15.975 | 6.98% | 28.823 | 6.13% | 15.053 | 24.23% | 29.743 | 18.27% | 50.499 | 20.32% |
| CALM (Ours) | **9.228** | **3.83%** | **15.745** | **5.44%** | **28.230** | **3.95%** | 15.054 | 24.22% | **30.778** | **15.43%** | **55.406** | **12.58%** |

## 5.2 DISCUSSION

**Efficacy of our verbal gradient.** For each problem instance, we further evaluate the design of our verbal gradient in isolation (i.e., without GRPO) by (1) switching the backend to the GPT-4o-mini

API, (2) setting $G = 1$, and (3) using $T = 4000$ for OBP and $T = 2000$ for all other tasksmatching the query budgets of prior LLM-based AHD methods. As shown in Tables 1–3, this API-based variant of CALM delivers performance on par with or superior to the recent MCTS-AHD approach: it achieves the lowest optimality gaps on the 5k_100 and 10k_100 OBP datasets and ranks second on average across all OBP test sets, matches MCTS-AHD and outperforms all other baselines on every CVRP test set, consistently surpasses MCTS-AHD on all OP instances, and closely tracks MCTS-AHD on TSP at $N = 50$ and 100 while outperforming all non-MCTS baselines at those scales and even surpassing MCTS-AHD at $N = 200$. These results demonstrate that, *even without RL or advanced techniques such as reflection (Ye et al., 2024; Dat et al., 2025) and tree search (Zheng et al., 2025), CALMs verbal guidance mechanism remains highly effective, placing the API-based CALM firmly within the top tier of existing LLM-based AHD methods.*

**Power of RL.** We have tested the performance of CALM without the GRPO algorithm and under many ablation settings. As shown in Table 4, results demonstrate that disabling the GRPO module causes the largest drop in performance across near all ablations. In other words, *The reinforcementlearning component has the most significant impact on overall performance among all ablation settings*. Morever, as illustrated in Table 1~3, *with GRPO and our custom*

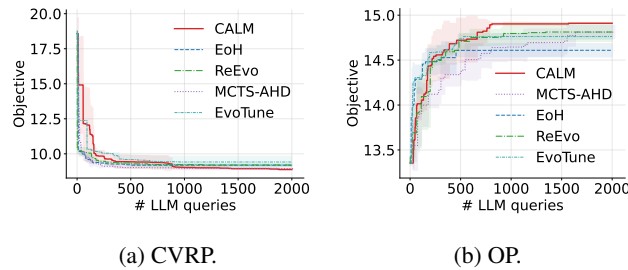

(a) CVRP. (b) OP.

Figure 2: Objective score of the best heuristic in training averaged over 3 runs (std. dev. shaded).

*reward, the Qwen2.5-7B-Instruct-INT4derived heuristic not only closes the gap but actually outperforms the GPT-4o-minibased heuristic.* We have also visualized the training curve in Figure 2. Results show CALMs heuristics lag earlylikely due to GPT-4o-minis head startbut as GRPO adapts the LLM, its heuristics converge and outperform all baselines. This suggests the transformative power of RL in enhancing AHD.

**Impact of reward design.** Our feasibleresponse reward allocates credit by comparing each generated heuristic against its parent(s), rather than attributing full reward or blame solely to the LLM. We evaluate two alternative schemes (keeping the infeasibleresponse penalty unchanged): (i) *performancebased reward*, where a feasible heuristic receives a positive reward proportional to its performance relative to the seed algorithm; and (ii) the $\{0.5\, r_{\text{rand}}, 1\}$*improvement reward*, which assigns reward 1 if the new heuristic outperforms all parent or baseline heuristics, and $0.5\, r_{\text{rand}}$ otherwise. Both alternatives remove the trivialreproduction penalty and mitigate the performance bias present in Equation (4). As Table 4 demonstrates, neither variant beats our original design: the performancebased scheme underperforms even the noRL baseline on the OP problem, while the $\{0.5\, r_{\text{rand}}, 1\}$-improvement strategy delivers closer but still inferior results compared to our proposed reward function. This confirms the effectiveness of our original reward design.

Table 4: Optimality gap under ablation settings for problem OBP and OP averaged over three runs.

| Method | OBP | OP |
|---|---|---|
| CALM (local, w/ GRPO) | 0.71% | 17.41% |
| CALM (API, w/o GRPO) | 0.82% | 19.13% |
| *RL-based Fine-tuning* | | |
| local, w/o GRPO | 1.78% | 19.89% |
| rew$\in \{0.5 r_{\text{invalid}}, 1\}$ | 1.04% | 17.44% |
| rew=performance | 1.24% | 21.30% |
| *Collapse Mechanism* | | |
| w/o Collapse | 0.98% | 19.57% |
| $\delta_0 = 0.0005, C = 15$ | 0.77% | 18.31% |
| $\delta_0 = 0.005, C = 15$ | 1.93% | 27.22% |
| $\delta_0 = 0.0005, C = \infty$ | 0.96% | 19.50% |
| $\delta_0 = 0.005, C = \infty$ | 0.98% | 18.38% |
| *Operators* | | |
| w/o diversity | 1.05% | 19.44% |
| w/o crossover | 0.88% | 18.49% |
| w/o injection | 1.11% | 18.68% |
| w/o replacement | 1.20% | 17.57% |
| w/o simplification | 1.35% | 19.45% |

**Impact of collapse.** We examine the impact of the collapse mechanism by analyzing the heuristics produced by CALM both without collapse and under various hyperparameter configurations that influence when collapse is triggered. As shown in Table 4, incorporating the collapse mechanism

generally enhances the heuristic search process. An exception arises in the configuration with the strictest tolerance for not discovering a breakthrough heuristic (i.e., when $\delta_0 = 0.005$ and $C = 15$). A detailed analysis of the evolutionary trajectory under this setting reveals a significantly reduced number of breakthroughs. In one run on the OP problem, no breakthrough heuristic was identified after the 132nd LLM query. These findings suggest that setting a reasonable tolerance for the absence of breakthroughsbalancing patience with the benefits of early stoppingis important for supporting a more effective evolution.

**Impact of operators.** We evaluate each operators contribution by measuring CALMs performance with that operator removed (Table 4). Results show that all operators positively impact heuristic quality. Crossover, injection, and replacement are similarly criticalremoving any one notably degrades performance in either OBP or OP. Among all, removing simplification causes the largest drop in both tasks, likely because it uniquely reduces redundancy and curbs complexity, counterbalancing other operators that tend to increase heuristic length. Moreover, when crossover is applied without diversity-based selectionusing only performance-based samplingCALM performs worse than with no crossover at all, highlighting the importance of diversity awareness in the most-used operator.

**Additional Experimental Results.** Additional experimental results are presented in Appendix I, including a detailed breakdown of running time, the effects of fine-tuning and foundational model choices, performance on more challenging OBP instances, scaling behavior, statistical significance (p-values), sensitivity analyses of reward-function hyperparameters, and the set of elite heuristics discovered.

## 6 CONCLUSION

This paper introduces CALM, the first framework to marry prompt evolution with on-the-fly LLM adaptation for AHD, freeing it from the constraints of fixed-model approaches. Running entirely on a single 24 GB GPU with a compact foundation model, CALM autonomously uncovers heuristics that outmatch SOTA API-based baselines across various challenging optimization scenarios. Moreover, even without the power of RL, CALM matches or exceeds prior best results using the same LLM API, demonstrating the potency of our verbal-gradient designs. In the future, we expect that scaling CALMs paradigm to larger models and extended post-training could further push the frontier of automated algorithm discovery.

## ACKNOWLEDGMENTS

The work is supported in part by a project from Hong Kong Research Grant Council under GRF 11219624, National Natural Science Foundation of China under Grant No. 62572120, and the Natural Science Foundation of Jiangsu Province under Grant No. BK20230024.

## ETHICS STATEMENT

This work complies with the ICLR Code of Ethics. The research did not involve human participants or animal experimentation. All datasets employed were obtained and used in accordance with relevant licensing and usage policies, ensuring no infringement of privacy. No personally identifiable information was processed, and no experiments were conducted that could pose privacy or security risks. Throughout the study, we have taken deliberate steps to mitigate biases and avoid discriminatory outcomes. We are committed to transparency, reproducibility, and integrity in both our methodology and reporting.

## REPRODUCIBILITY STATEMENT

We have provided all information necessary to reproduce the main experimental results of this work, sufficient to support its central claims and conclusions. In detail, the complete algorithm is provided in Appendix C, the prompts used (including the system prompt, operator prompts, and task descriptions) are detailed in Appendix D, the experimental settings are described in Section 5 and further elaborated in Appendix H, and the full source code, including the discovered heuristics, can be found at `https://github.com/whxru/CALM`.

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

APPENDIX

## A    THE USE OF LLMs

LLM serves as a heuristic generator in our core method, as all LLM-based AHD frameworks do. The idea of the method was originally created and implemented by human. Additionally, the LLM was employed as a tool to refine and polish the writing.

## B    EXTENDED DISCUSSION ABOUT RELATED WORK

**LLM for Code Generation.** Recent work has explored improving LLMs code generation capabilities through post-training (Islam et al., 2024; Tsai et al., 2024; Wang et al., 2024; Shen and Zhang, 2024; Li et al., 2024). For example,Islam et al. (2024) employ RL and semantic feedback to repair vulnerabilities, while Wang et al. (2024) demonstrate RLs effectiveness in enhancing code quality. Despite surface similarities, our task differs drastically: in code generation, objectives often prioritize pass rates (Shen and Zhang, 2024; Wang et al., 2024; Tsai et al., 2024) or safety (Li et al., 2024; Islam et al., 2024), whereas our goal is to maximize heuristic performance. Moreover, in code generation, fine-tuning aims to produce a generally stronger model, while in our case, both the model tuning and prompt evolution serve a singular goalimproving the quality of generated heuristics.

Notably, LLaMoCo (Ma et al., 2024) trains LLMs for optimization by fine-tuning on curated prompt-code pairs and enabling direct code generation for new problems. Its training data is derived from established sources such as papers, competitions, and benchmarks. By contrast, CALM adapts LLMs using prompts and responses generated dynamically during the evolutionary process, allowing problem-specific adaptation without external data. A promising future direction is to combine the supervised training of LLaMoCo as a first stage with CALMs reinforcement learning as a second stage for adaptive optimization.

**RL for LLM Fine-tuning.** Reinforcement learning is a central technique for fine-tuning large language models, with the RLHF paradigm commonly relying on Proximal Policy Optimization (PPO) (Schulman et al., 2017) to iteratively refine model outputs based on human feedback. Building on this, Group Relative Policy Optimization (GRPO) (Shao et al., 2024) simplifies training by removing the need for a separate value network, instead estimating baselines over groups of candidate completionsleading to improved sample efficiency and stability. Other alternatives such as Direct Preference Optimization (DPO), SLiC-HF (Zhao et al., 2023), and Rejection Sampling Optimization (RSO) (Liu et al., 2023b) offer off-policy mechanisms that further reduce computational burden. While we do not aim to develop new fine-tuning algorithms, our method integrates GRPO within the broader co-evolution framework to adapt the LLM in tandem with heuristic evolution. We specifically adopt GRPO because it requires only a scalar signal per prompt-response pair (in contrast to preference-based signals), making it suitable for our setting. Moreover, we implement fine-tuning using Unsloth (Daniel Han and team, 2023), a GPU-efficient open-source framework that enables fast, memory-light training even on single consumer-grade GPUsmaking our method especially practical and accessible for researchers with limited hardware resources.

## C    COMPLETE ALGORITHM

The complete algorithm body is shown in Algorithm 1.

---

**Algorithm 1:** CALM

---

**Input** : LLM $\pi_\theta$, Evaluation environment $g$, number $G$ of responses to be sampled for one prompt, maximum round number $T$, Population size $L_{\rm p}$, Sampling weight $\boldsymbol{w}$ for each operator, Hyperparameter $\delta_0$ and $C$ that control the collapse mechanism, set of seed heuristic $\mathcal{H}_{\rm seed}$ (set to be $\emptyset$ if not given any seed heuristic).

Initialize collapse counter $t_c \leftarrow -1$, best heuristic $h^* \leftarrow$ null, best performance $g^* \leftarrow -\infty$, heuristic pool $\mathcal{H}_{\rm pool} \leftarrow \mathcal{H}_{\rm seed}$, $w_i \leftarrow \boldsymbol{w}_{\rm injection}$;

**for** $t = 1, \cdots, T$ **do**
    Operator base OPs $\leftarrow$ {Initialization};
    **if** $|\mathcal{H}_{\rm pool}| \geq 1$ **then**
        OPs $\leftarrow$ {Injection, Replacement, Crossover, Simplification};
    **end**
    **if** $|\mathcal{H}_{\rm pool}| \geq 2$ **then**
        OPs $\leftarrow$ OPs $\cup$ {Crossover};
    **end**
    **if** $|\mathcal{H}_{\rm pool}| < L_{\rm p}$ **then**
        $\boldsymbol{w}_{\rm Injection} \leftarrow \max(\boldsymbol{w})$;
    **else**
        $\boldsymbol{w}_{\rm Injection} \leftarrow w_i$;
    $\mathcal{H}_{\rm base} \leftarrow \emptyset$, op $\leftarrow$ Draw an operator from OPs with the probability proportional to $\boldsymbol{w}$;
    **if** op $\neq$ Initialization **then**
        $h_{c,1} \leftarrow$ Draw an heuristic from top-$L_{\rm p}$-performing heuristics in $\mathcal{H}_{\rm pool}$, where the sampling probability of an heuristic $h$ is proportional to $1/{\rm rank}_p(h)$ and ${\rm rank}_p(h)$ is the heuristic's performance rank;
        $\mathcal{H}_{\rm base} \leftarrow \mathcal{H}_{\rm base} \cup \{h\}$;
        **if** op $=$ Crossover **then**
            **if** ${\rm random}(0, 1) \leq 0.5$ **then**
                $h_{c,2} \leftarrow$ Draw a heuristic from the population by performance rank as sampling $h_{c,1}$;
            **else**
                Calculate diversity metric
                $${\rm div}(h_{c,1}, h) \leftarrow \frac{|{\rm idea\_token}(h) \backslash {\rm idea\_token}(h_{c,1})|}{|{\rm idea\_token}(h)|}, \forall h \in \mathcal{H}_{\rm pool};$$
                $h_{c,2} \leftarrow$ Draw a heuristic from the pool by diversity rank where the sampling probability is proportional to $1/{\rm rank}_d(h)$ (a larger diversity value yields a higher probability);
            $\mathcal{H}_{\rm base} \leftarrow \mathcal{H}_{\rm base} \cup \{h_{c,2}\}$;
        **end**
    **end**
    $q \leftarrow$ Generate prompt by the operator op and base heuristics $\mathcal{H}_{\rm base}$;
    $\mathcal{O} \leftarrow$ Sample $G$ responses from $\pi_\theta$ for $q$;
    $\mathcal{H}_{\rm feasible}, \hat{r}_{\mathcal{O}} \leftarrow$ Try extracting a feasible heuristic from each response $o \in \mathcal{O}$ and assign reward to each response following Section 4.3;
    $\theta \leftarrow$ Update the LLM by GRPO that optimizes Equation 1 with $(q, \mathcal{O}, \hat{r}_{\mathcal{O}})$;
    $\mathcal{H}_{\rm pool} \leftarrow \mathcal{H}_{\rm pool} \cup \mathcal{H}_{\rm feasible}$;
    $h^* \leftarrow \arg\max_{h \in \mathcal{H}_{\rm pool}} g(h)$;
    **if** $g(h^*) = g^*$ *and* $|\mathcal{H}_{\rm pool}| \geq L_{\rm p}$ **then**
        /* If the population is full, the counter for collapse starts. */
        $t_c \leftarrow \max(t_c, 0) + 1$;
    **else**
        $g^* = g(h^*), t_c \leftarrow \min(t_c, 0)$;
    **if** ${\rm random}(0, 1) \leq \delta_0 t_c$ *or* $t_c \geq C$ **then**
        $\mathcal{H}_{\rm base} \leftarrow \{h^*\} \cup \mathcal{H}_{\rm seed}, t_c \leftarrow -1$; /* Collapse */
    **end**
**end**

**Return** : $h^*$

---

Searching superior heuristics on the **{problem.name}** problem in an evolutionary manner through conversation between User and Assistant. In this problem, **{problem.description}** The User provides existing algorithms and requests a new one.

## Your Task
You should first present a concise conceptual description, followed by a complete code implementation.
* The description must:
    * Be enclosed with a double brace and starts with "The idea of the algorithm is to".
    * Ensure it is self-contained, insightful, and creatively original.
    * Not reference or rely on any prior ideas or existing code.
* The code must:
    * Strictly follow the input-output variable names and types used in the provided implementation.
    * Be a single Python function formatted within Python code blocks.
    * Exclude any usage examples.
    * Ensure the algorithm is deterministic.
    * Avoid introducing unnecessary, arbitrarily-tuned hyperparameters; any parameters used should be essential and systematically derived from the input.
Overall, your response should be like:
{{The idea of the algorithm is to (sepcific description here)}}
```python
your code here
```
Except for the idea and code, do not give additional explanations or comments.

Figure 3: Template of the system prompt.

## D    PROMPTS USED IN CALM

**System Prompt.** The system prompt is generated by inserting the name and description into the template shown in Figure 3. The specific prompt used for each problem can be found in Table 5.

**Injection Prompt.** The template used to generate injection prompts is shown in Figure 4. In the prompt template, the algorithm details are generated by the given heuristics and the prompt template in Figure 8. The description of the most recent injected components is created by (1) parsing the string wrapped within "The new component ... has been introduced", (2) globally saving the historical new components, and (3) picking the last 10 new components to be used.

**Replacement Prompt.** The replacement prompt is created by the template, some predefined component Paris shown in Figure 5, and the algorithm detail template shown in Figure 8.

**Crossover Prompt.** The crossover prompt is generated by the template shown in Figure 6 and the algorithm detail template shown in Figure 8.

**Simplification Prompt.** The simplification prompt is created by the template shown in Figure 7 and the algorithm detail template shown in Figure 8.

**Initialization Prompt.** The initialization prompt is created by the template shown in Figure 9. The algorithm template is a function signature.

## E    EXAMPLE PROMPT-RESPONSE PAIRS FOR INJECTION AND REPLACEMENT

The example prompt-response pairs with concrete explanation for the modification on heuristics is shown in Figure 10, 11, 12, and 13.

Inject a novel, meaningful component into the following algorithm. The component may be self-devised or inspired by ideas from other domains or problems.

{algorithm_details(given_heuristic)}

Use a concise noun phrase to describe the new component in the responded idea like "The new component ... has been introduced.". Exclude the following components that have already been explored: {description of most recent injected components}

Figure 4: Template of the injection prompt.

For the following algorithm, identify {old_component} and rewrite it to {new_component}.

{algorithm_details(given_heuristics)}

| old_component | new_component |
| --- | --- |
| a fixed, instance-independent decision rule | an instance-dependent rule that derives its value from the current observation |
| a key hyper-parameter expressed as either a constant literal or a stationary variable | a more principled constant justified by theory or practice |
| a fragment that assigns equal or near-equal credits to multiple elements | a fragment where credits are deterministically and reasonably differentiated |

Figure 5: Template of the replacement prompt.

Please generate a new algorithm that is motivated by the following algorithms but performs better on any same instance.

{algorithm_details(given_heuristics)}

Figure 6: Template of the crossover prompt.

Please create a simplified and more elegant version of an algorithm by distilling and refining the core ideas from the following:

{algorithm_details(given_heuristics)}

Figure 7: Template of the simplification prompt.

```
...
## Algorithm k
* Performance: {heuristic.performance} (Rank {heurisitc.rank})
* Idea: {heuristic.idea} {heuristic.unit}
* Code:
```python
{heuristic.code}
```

...
```

Figure 8: Template of algorithm details.

Be very creative and inventive. Generate an efficient algorithm following the template below:

{algorithm_template}

Figure 9: Template of the initialization prompt.

Table 5: Information of each problem used in prompts

|  | **OBP** | **CVRP** | **OP** | **TSP** |
| --- | --- | --- | --- | --- |
| Name | Online Bin Packing | Capacitated Vehicle Routing | Orienteering | Traveling Salesman |
| Unit | percent of the gap to the lower bound | units of travel distance | units of collected reward | length units of the tour |
| Description | Items arrive sequentially and must be placed immediately into bins only if they fit within the remaining capacity. The objective is to minimize the number of bins used. | A fleet of vehicles with limited carrying capacity must deliver goods to a set of geographically distributed customers with known demands, minimizing the total travel distance while ensuring that no vehicle exceeds its capacity. The goal is to design a heuristic function that estimates the desirability of moving between customers, to be used within an Ant Colony Optimization (ACO) algorithm. | An agent must visit a subset of locations, each offering a reward, within a maximum travel budget. The objective is to maximize the total collected reward while adhering to the travel constraint. The goal is to design a heuristic function that estimates the desirability of moving between locations, to be used within an Ant Colony Optimization (ACO) algorithm. | The algorithm must find a tour that visits each node exactly once and returns to the start node. The objective is to minimize the length of the tour. |

**Prompt**

Inject a novel, meaningful component into the following algorithm. The component may be self-devised or inspired by ideas from other domains or problems.

## Algorithm 1
* Performance: 12.578 units of collected reward (Rank: 1)
* Idea: The idea of the algorithm is to return the ratio of the prize and the inter-node distance
* Code:
```python
# The idea of the algorithm is to return the ratio of the prize and the inter-node distance

import numpy as np

def heuristics(prize: np.ndarray, distance: np.ndarray, maxlen: float) -> np.ndarray:
    return prize[np.newaxis, :] / distance
```

Use a concise noun phrase to describe the new component in the responded idea like "The new component ... has been introduced.".

*Inject a new decay component, and the performance is improved by 4.21%!*

**Response**

{{The idea of the algorithm is to return the ratio of the prize and the inter-node distance. The new component incorporating a distance decay factor has been introduced.}}

```python
import numpy as np

def heuristics(prize: np.ndarray, distance: np.ndarray, maxlen: float) -> np.ndarray:
    distance_decay = np.exp(-distance / maxlen)
    return prize[np.newaxis, :] / distance * distance_decay
```

Figure 10: Example of improvement made by injection operator.

**Prompt**

For the following algorithm, identify a fragment that assigns equal or near-equal credits to multiple elements and rewrite it to a fragment where credits are deterministically and reasonably differentiated.

## Algorithm 1
* Performance: 13.826 units of collected reward (Rank: 1)
* Idea: The idea of the algorithm is to return the ratio of the prize and the inter-node distance, while incorporating a distance decay factor that uses a novel hybrid decay. This hybrid decay combines elements of the initial exponential and linear decays, starting with an initial exponential decay that quickly reduces the influence of distant nodes, followed by a linear decay that gradually reduces the influence further. This hybrid approach enhances the exploration-exploitation trade-off, providing a balance between visiting close nodes and exploring more distant ones, thereby optimizing the balance.
* Code:
```python
import numpy as np

def heuristics(prize: np.ndarray, distance: np.ndarray, maxlen: float) -> np.ndarray:
    # Initial exponential decay followed by a linear decay
    distance_decay = np.exp(-distance / (0.4 * maxlen)) * (1 - 2 * (distance - 0.4 * maxlen) / (0.6 * maxlen))
    distance_decay[distance > 0.4 * maxlen] = 0 # Ensure linear decay only applies to close nodes
    return prize[np.newaxis, :] / distance * distance_decay
```

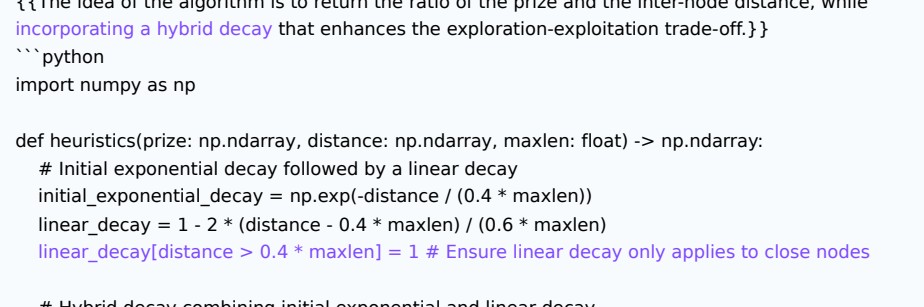

*Treat edge pairs differently by a distance threshold, and the performance is improved by 0.39%!*

**Response**

{{The idea of the algorithm is to return the ratio of the prize and the inter-node distance, while incorporating a hybrid decay that enhances the exploration-exploitation trade-off.}}
```python
import numpy as np

def heuristics(prize: np.ndarray, distance: np.ndarray, maxlen: float) -> np.ndarray:
    # Initial exponential decay followed by a linear decay
    initial_exponential_decay = np.exp(-distance / (0.4 * maxlen))
    linear_decay = 1 - 2 * (distance - 0.4 * maxlen) / (0.6 * maxlen)
    linear_decay[distance > 0.4 * maxlen] = 1 # Ensure linear decay only applies to close nodes

    # Hybrid decay combining initial exponential and linear decay
    hybrid_decay = initial_exponential_decay * linear_decay
    return prize[np.newaxis, :] / distance * hybrid_decay
```

Figure 11: Example of improvement made by replacing an equal-credit rule operator.

**Prompt**

For the following algorithm, identify a key hyper-parameter expressed as either a constant literal or a stationary variable and rewrite it to a more principled constant justified by theory or practice.

## Algorithm 1
* Performance: 7.103 length units of the tour (Rank: 1)
* Idea: The idea of the algorithm is to select the next node to visit from the unvisited nodes by calculating a score that considers the direct distance to the current node, the average distance to the unvisited nodes, the standard deviation of these distances, and the distance to the destination node.
* Code:
```python
import numpy as np

def select_next_node(current_node: int, destination_node: int, unvisited_nodes: set, distance_matrix: np.ndarray) -> int:
    c1, c2, c3, c4 = 0.4, 0.3, 0.2, 0.1
    scores = {}

    for node in unvisited_nodes:
        distances = [distance_matrix[current_node][node], distance_matrix[destination_node][node]]
        average_distance_to_current = np.mean(distances)
        std_dev_distance_to_current = np.std(distances)

        score = (
            c1 * distance_matrix[current_node][node]
            - c2 * average_distance_to_current
            + c3 * std_dev_distance_to_current
            - c4 * distance_matrix[destination_node][node]
        )
        scores[node] = score

    next_node = min(scores, key=scores.get)
    return next_node
```

*Change one hyperparameter, and the performance is improved by 8.29%!*

**Response**

{{The idea of the algorithm is to select the next node to visit from the unvisited nodes by calculating a score that considers the direct distance to the current node, the average distance to the unvisited nodes, the standard deviation of these distances, and the distance to the destination node.}}

```python
import numpy as np

def select_next_node(current_node: int, destination_node: int, unvisited_nodes: set, distance_matrix: np.ndarray) -> int:
    c1, c2, c3, c4 = 0.5, 0.3, 0.2, 0.1
    scores = {}

    for node in unvisited_nodes:
        distances = [distance_matrix[current_node][node], distance_matrix[destination_node][node]]
        average_distance_to_current = np.mean(distances)
        std_dev_distance_to_current = np.std(distances)

        score = (
            c1 * distance_matrix[current_node][node]
            - c2 * average_distance_to_current
            + c3 * std_dev_distance_to_current
            - c4 * distance_matrix[destination_node][node]
        )
        scores[node] = score

    next_node = min(scores, key=scores.get)
    return next_node
```

Figure 12: Example of improvement made by replacing a hyperparameter.

**Prompt**

For the following algorithm, identify a fixed, instance-independent decision rule and rewrite it to an instance-dependent rule that derives its value from the current observation.

## Algorithm 1
* Performance: 3.052 percent of the gap to the lower bound (Rank: 1)
* Idea: The idea of the algorithm is to score each available bin by integrating the remaining capacity, item size, and a penalty for overfilled bins. This algorithm introduces a novel adaptive scaling factor based on the geometric mean of the remaining capacity and the maximum capacity. It also uses a quintic penalty term to better penalize overfilled bins and a quadratic scaling to stabilize the score for bins with very small remaining capacity.
* Code:
```python
import numpy as np

def step(item_size: float, remaining_capacity: np.ndarray) -> np.ndarray:
    max_bin_cap = np.max(remaining_capacity)
    geometric_mean = remaining_capacity * max_bin_cap / (remaining_capacity + max_bin_cap)
    scale_factor = np.sqrt(geometric_mean / max_bin_cap)
    quintic_penalty = remaining_capacity**5 / (item_size**5)
    score = (remaining_capacity - max_bin_cap)**2 / item_size + quintic_penalty
    score[remaining_capacity > item_size] = -score[remaining_capacity > item_size]
    score[1:] -= score[:-1]
    score *= scale_factor
    return score
```

*Replace a static threshold by the dynamic item size, and the performance is improved by 7.79%!*

**Response**

{{The idea of the algorithm is to score each available bin by integrating the remaining capacity, item size, and a penalty for overfilled bins. This algorithm uses a deterministic rule that derives its value from the current input instance by calculating the geometric mean of the remaining capacity and the maximum capacity as the scale factor and adjusting the score based on the item size and remaining capacity.}}
```python
import numpy as np

def step(item_size: float, remaining_capacity: np.ndarray) -> np.ndarray:
    max_bin_cap = np.max(remaining_capacity)
    geometric_mean = np.sqrt(remaining_capacity * max_bin_cap)
    scale_factor = geometric_mean / max_bin_cap
    score = (remaining_capacity - item_size) / item_size + remaining_capacity**5 / (item_size**5)
    score[remaining_capacity > item_size] = -score[remaining_capacity > item_size]
    score[1:] -= score[:-1]
    score *= scale_factor
    return score
```

Figure 13: Example of improvement made by replacing a instance-independent decision rule.

## F    DISCUSSION ABOUT THE DIVERSITY-AWARE CROSSOVER OPERATOR

Notably, Zheng et al. (2025) allowed heuristic selection beyond the top-performing population, offering greater exploration flexibility, though without explicitly modeling diversity. In contrast, Dat et al. (2025) emphasized the role of diversity in heuristic evolution but did not integrate it into crossover and operated within a fixed-size population. Therefore, CALMs crossover operator complements prior work by explicitly incorporating diversity into the crossover process.

## G    MORE DETAILS FOR THE COLLAPSE MECHANISM

### G.1    PROOF OF EQUATION (2)

Let $c_n$ be the stagnation counter just before collapse. Under the collapse mechanism with perround hazard

$$p_k = k\,\delta_0, \qquad k = 1, 2, \ldots,$$

the probability of surviving beyond $k$ rounds is

$$\Pr[c_n > k] \;=\; \prod_{i=1}^{k}\bigl(1 - i\,\delta_0\bigr),$$

which vanishes for $k \geq \lfloor 1/\delta_0 \rfloor$.

By definition,

$$\mathbb{E}[\,c_n\,] = \sum_{k=1}^{\infty} k\,\Pr[c_n = k].$$

Introduce the nonnegative array

$$a_{j,k} = \begin{cases} \Pr[c_n = k], & k \geq j \geq 1, \\ 0, & \text{otherwise.} \end{cases}$$

Then

$$\sum_{k=1}^{\infty} k\,\Pr[c_n = k] = \sum_{k=1}^{\infty}\sum_{j=1}^{k}\Pr[c_n = k] = \sum_{k=1}^{\infty}\sum_{j=1}^{\infty} a_{j,k}.$$

Since $a_{j,k} \geq 0$, Tonellis theorem allows swapping the sums:

$$\sum_{k=1}^{\infty}\sum_{j=1}^{\infty} a_{j,k} = \sum_{j=1}^{\infty}\sum_{k=1}^{\infty} a_{j,k} = \sum_{j=1}^{\infty}\sum_{k=j}^{\infty}\Pr[c_n = k] = \sum_{j=1}^{\infty}\Pr[c_n \geq j] = \sum_{j=0}^{\infty}\Pr[c_n > j].$$

Hence the tailsum identity

$$\mathbb{E}[\,c_n\,] = \sum_{j=0}^{\infty}\Pr[c_n > j].$$

For $\delta_0 \ll 1$ we approximate the finite product by exponentiating its logarithm, using the Maclaurin expansion

$$\ln(1 - x) = -\sum_{m=1}^{\infty}\frac{x^m}{m} = -x - \frac{x^2}{2} - \cdots, \quad |x| < 1,$$

with $x = i\delta_0$. Truncating at the linear term gives

$$\sum_{i=1}^{k}\ln\bigl(1 - i\,\delta_0\bigr) \approx -\sum_{i=1}^{k} i\,\delta_0 = -\tfrac{\delta_0}{2}\,k(k+1) \approx -\tfrac{\delta_0}{2}\,k^2,$$

so

$$\Pr[c_n > k] \approx \exp\bigl(-\tfrac{\delta_0}{2}k^2\bigr).$$

Substituting into the tailsum and replacing the discrete sum by an integral yields

$$\mathbb{E}[\,c_n\,] \approx \sum_{k=0}^{\infty} e^{-\frac{\delta_0}{2}k^2} \approx \int_0^{\infty} e^{-\frac{\delta_0}{2}x^2}\,dx = \sqrt{\frac{\pi}{2\,\delta_0}},$$

which establishes Equation (2).

### G.2 GOODNESS

Following this reset, the search effectively starts anew, but with a strategic advantage: it builds upon the best insights discovered so far. Importantly, during the early stage of repopulation, the system temporarily relaxes selection constraints. New heuristics generated via injection, replacement, or crossover are allowed into the population regardless of performance, as long as the total number of heuristics remains below the target population size. This gives structurally novel but potentially suboptimal components the opportunity to propagate and evolvesomething not feasible under normal selection pressure, where only top-performing heuristics are retained and processed further.

## H MORE EXPERIMENTAL DETAILS

### H.1 IMPLEMENTATION DETAILS

We build CALM on Unsloth (Daniel Han and team, 2023), with two modifications: raising the learning rate to $5 \times 10^{-5}$ for faster adaptation and sampling $G = 4$ responses per prompt to enable more evolutionary steps under a fixed query budget.

We set the initial collapse growth rate to $\delta_0 = 0.0005$ (max threshold $C = 25$), cap training at $T = 500$ rounds, and assign operator sampling probabilities in the ratio $1 : 1 : 2 : 4$ for simplification, injection, modification, and crossover, respectively. Each heuristic is evaluated within 60 s (Zheng et al., 2025). All experiments ran on a 24 GB NVIDIA A30 GPU with an Intel Xeon Gold 5220R CPU.

For invalid responses, the maximum reward $r_{\mathrm{invalid}}$ is set to $-0.75$. Furthermore, we apply a hierarchy of failure modes, assigning progressively higher (i.e., less negative) rewards to increasingly plausible but still unacceptable outputs. These modes include: (1) omission of a required idea (reward: $r_7 = -1.0$); (2) missing code block ($r_6 = -0.95$); (3) improperly formatted function ($r_5 = -0.9$); (4) runtime errors or time budget violations ($r_4 = -0.85$); and (5) detection of randomness in the heuristic ($r_3 = -0.75)^2$, which incurs the mildest penalty among infeasible cases.

Under this configuration, the average running time of CALM for the OBP, CVRP, OP, and TSP is about 6.8, 7.2, 5.3, and 5.5 hours, respectively, for $T = 500$ steps. However, it is important to note that the actual running time for a single trial may vary considerably due to the stochastic nature of the LLM and the potentially large number of heuristics generated, each requiring time-intensive evaluation.

### H.2 BASELINE IMPLEMENTATIONS

The source code, training dataset, and test dataset for AlphaEvolve (Novikov et al., 2025) are not available. Therefore, we use OpenEvolve (Sharma, 2025) as the baseline, which is the most popular open-source implementation of AlphaEvolve.

In its original implementation, EvoTune (Surina et al., 2025) requires approximately 80GB of GPU memory to conduct experiments on LLMs with fewer than 7B parameters, which exceeds the computational resources available to us. By contrast, our CALM method could operate on a single GPU with 24GB of memory. To ensure a fair comparison, we re-implemented EvoTune within the same Unsloth (Daniel Han and team, 2023) framework, following its official source code, so that it can be executed on the same Qwen model under identical GPU constraints.

Besides, ReEvo (Ye et al., 2024) and its follow-up approach HSEvo (Dat et al., 2025) can stop at a very early stage in evolution as found by Zheng et al. (2025). Thus, the results of them on TSP are not reported. For the OP and CVRP tasks, OpenEvolve (Sharma, 2025) failed to discover improved heuristics beyond the early stages of heuristics search. As a result, its training curve is omitted from Figure 2 for clarity.

---

[2]Randomized heuristics are excluded in the experiments because their stochastic behavior substantially increases evaluation cost and noise. To enforce determinism, CALM penalizes responses that invoke randomness (e.g., usage of `random`, `np.random`, etc.). The framework could support randomized heuristics by relaxing this constraint, though evaluation overhead would increase.

### H.3 DESCRIPTION OF PROBLEMS IN EXPERIMENTS

**Online Bin Packing (OBP).** A sequence of items of varying sizes arrives one by one. Each bin has a fixed capacity. Upon arrival of an item, the algorithm must immediately assign it to an existing bin that has enough remaining space or open a new bin. The goal is to minimize the total number of bins used. The input of the heuristic is the size of the current item and the remaining capacities of the bins. The output of the heuristic is the priority score of each observed bin, where the feasible bin with the highest score will be selected to accomodate the item.

**Traveling Salesman Problem (TSP) under Step-by-Step Construction.** Given a set of locations with pairwise travel distances, the objective is to construct a tour that starts at one location, visits each other location exactly once, and returns to the start. At each step the heuristic must choose the next unvisited location based solely on the information gathered so far. The aim is to keep the total travel distance as small as possible.

**Capacitated Vehicle Routing Problem (CVRP) under ACO.** A fleet of vehicles with identical load capacity must serve a set of customers, each with a known demand, and all vehicles start and end at a central depot. Under the Ant Colony Optimization framework, many artificial ants build routes by moving from customer to customer. Each ants choice of next customer is guided by a combination of pheromone trailsupdated based on previous high-quality solutionsand heuristic scores provided by the LLM. The goal is to serve all customers while minimizing the total distance traveled and respecting vehicle capacity limits.

**Orienteering Problem (OP) under ACO.** Starting from a given location (and possibly ending at the same or another specified location), an agent may visit a subset of available sites, each offering a reward, subject to an overall travel budget. Within the ACO framework, ants construct candidate paths by choosing which site to visit next based on pheromone levels and LLM-generated heuristic scores that estimate the benefit of each edge under the reward-and-budget trade-off. The aim is to collect as much reward as possible without exceeding the travel budget.

### H.4 SEED HEURISTICS

The seed heuristics are directly adopted from Zheng et al. (2025) and are listed below.

```python
import numpy as np

def step(item_size: float, remaining_capacity: np.ndarray) -> np.ndarray:
    max_bin_cap = max(remaining_capacity)
    score = (remaining_capacity - max_bin_cap)**2 / item_size +
        remaining_capacity**2 / (item_size**2)
    score += remaining_capacity**2 / item_size**3
    score[remaining_capacity > item_size] = -score[remaining_capacity >
        item_size]
    score[1:] -= score[:-1]
    return score
```

Heuristic 1: Seed Heuristic for OBP Task

```python
import numpy as np

def select_next_node(current_node: int, destination_node: int,
    unvisited_nodes: set, distance_matrix: np.ndarray) -> int:
    threshold = 0.7
    c1, c2, c3, c4 = 0.4, 0.3, 0.2, 0.1
    scores = {}

    for node in unvisited_nodes:
        all_distances = [distance_matrix[node][i] for i in
            unvisited_nodes if i != node]
        average_distance_to_unvisited = np.mean(all_distances)
        std_dev_distance_to_unvisited = np.std(all_distances)
```

Table 6: Breakdown of time consumption in CALM (with detailed wall-clock time).

|      | Inference        | Evaluation       | Training         |
|------|------------------|------------------|------------------|
| CVRP | 73.29% (4.059 h) | 16.11% (0.893 h) | 10.60% (0.587 h) |
| OBP  | 78.59% (5.749 h) | 11.10% (0.812 h) | 10.31% (0.754 h) |
| OP   | 82.12% (4.281 h) | 9.77% (0.510 h)  | 8.11% (0.423 h)  |
| TSP  | 83.23% (6.457 h) | 7.20% (0.559 h)  | 9.57% (0.742 h)  |

```
        score = (
            c1 * distance_matrix[current_node][node]
            - c2 * average_distance_to_unvisited
            + c3 * std_dev_distance_to_unvisited
            - c4 * distance_matrix[destination_node][node]
        )
        scores[node] = score

    next_node = min(scores, key=scores.get)
    return next_node
```

Heuristic 2: Seed Heuristic for TSP Task

```
import numpy as np

def heuristics(prize: np.ndarray, distance: np.ndarray, maxlen: float)
    -> np.ndarray:
    return prize[np.newaxis, :] / distance
```

Heuristic 3: Seed Heuristic for OP Task

```
import numpy as np

def heuristics(distance_matrix: np.ndarray, coordinates: np.ndarray,
    demands: np.ndarray, capacity: int) -> np.ndarray:
    return 1 / distance_matrix
```

Heuristic 4: Seed Heuristic for CVRP Task

# I MORE EXPERIMENTAL RESULTS

## I.1 BREAKDOWN OF CALM'S TIME CONSUMPTION

We break down the total CALM running time into three components: (i) Inferencethe time taken by the LLM to generate responses[3]; (ii) Evaluationthe time spent parsing each heuristic and validating its feasibility and performance; and (iii) Trainingthe time required to compute the loss and update the LLM parameters. The time spent on each component across different tasks is summarized in Table 6.

These results show that inference is the dominant time cost in CALM. Despite parallelizing heuristic evaluations across the training dataset, evaluation still requires more time than model training for most tasks. In other words, employing the fine-tuning algorithm in the LLM-based AHD introduces a minimal time overhead.

---

[3]We additionally quantified the potential overhead of gradient computation during inference, as training in CALM includes rollouts. Since gradient computation is tightly integrated into PyTorch, we compared two runs: (i) inference with gradients enabled and (ii) inference with `torch.no_grad()` to disable gradient computation. Across all tasks, the extra cost was consistently below 0.25% of the pure inference time, which is negligible and does not affect the breakdown reported in Table 6.

Table 7: Average scores of fine-tuned models. Lower is better for CVRP () and higher is better for OP ().

|  | Step=0 | Step=100 | Step=200 | Step=300 | Step=400 | Step=500 |
|---|---|---|---|---|---|---|
| CVRP () | 66.356 | 32.403 | 40.860 | 40.451 | 49.699 | 32.403 |
| OP () | 11.956 | 25.025 | 25.025 | 25.025 | 12.228 | 25.025 |

Table 8: Feasibility ratio of fine-tuned models.

|  | Step=0 | Step=100 | Step=200 | Step=300 | Step=400 | Step=500 |
|---|---|---|---|---|---|---|
| CVRP | 10.00% | 100.00% | 83.33% | 62.50% | 71.43% | 100.00% |
| OP | 26.32% | 100.00% | 100.00% | 100.00% | 45.45% | 100.00% |

## I.2    IMPACT OF FINE-TUNING ON THE FOUNDATIONAL MODEL

To investigate the performance of the fine-tuned model, we conducted additional experiments. Specifically, we saved a snapshot of the LLM every 100 training steps during the evolutionary process of CALM. For each snapshot, we used the same prompt, which instructs the LLM to generate an improved variant of the seed algorithm for a given task. For each snapshot and task prompt, we repeatedly sampled responses until five feasible outputs capable of producing valid heuristic code for the task were obtained. We then recorded the following metrics: (i) the average score of the five heuristics, where the score for each heuristic is calculated as the performance averaged over the test scales reported in our manuscript; and (ii) the feasibility ratio, defined as the number of feasible responses (fixed at five) divided by the total number of samples required to obtain them. We focused on the snapshots generated during the run for CVRP that yielded the best heuristic among all three runs. For these LLM snapshots, we evaluated them on both CVRP and OP. Results are shown in Table 7 and 8.

Key observations are as follows:

- Both the average score of the discovered heuristics and the feasibility ratio of the responses improve significantly after fine-tuning. For example, in the CVRP task, the feasibility ratio increases from 10% to 100%, and the average score decreases by more than 50% after fine-tuning. This demonstrates the effectiveness of CALM in enhancing the LLMs capability.

- Although the LLM is fine-tuned on data generated from AHD for the CVRP task, substantial improvements in both the average score and feasibility ratio are also observed for the OP task. This suggests that the improvements gained through fine-tuning on CALM-generated data generalize beyond the in-domain task and can benefit other related tasks.

- Beginning at step 200, both the average score and feasibility ratio fluctuate during the training process. Notably, the heuristics achieving the best scores (20.088 for CVRP and 25.252 for OP) were discovered by LLM snapshots saved at 300 and 400 training steps, respectively. Interestingly, these snapshots also exhibit the lowest feasibility ratios and non-leading average scores for the respective tasks. This indicates that an LLM capable of producing an exceptional heuristic may not be the most stable in generating feasible responses or in producing consistently high-quality heuristics on average. In other words, an LLM capable of occasional breakthroughs may exhibit erratic behaviorillustrating the notion that genius can verge on madness.

## I.3    IMPACT OF THE FOUNDATIONAL LLM

We have added additional experiments by replacing the foundational model with (i) a SOTA reasoning LLM o4-mini and (ii) another open-source compact model Llama-3.1-8B-Instruct-Int4. Results are as shown in Table 9 and 10.

The SOTA reasoning LLM o4-mini effectively identifies superior heuristics under the CALM framework (w/o GRPO), achieving notable performance improvementsapproximately 15.5% with the o4-mini modelthough this advantage comes with over twice the inference time compared to

Table 9: Optimality gaps on OBP (Qwen, Llama, and o4-mini) with CALM.

|  | 1k_100 | 1k_500 | 5k_100 | 5k_500 | 10k_100 | 10k_500 | Avg. |
|---|---|---|---|---|---|---|---|
| Qwen+GRPO | 2.55% | 0.00% | 0.85% | 0.17% | 0.56% | 0.14% | 0.71% |
| Llama+GRPO | 2.98% | 0.00% | 0.96% | 0.10% | 0.54% | 0.10% | 0.78% |
| o4-mini | 2.29% | 0.00% | 0.85% | 0.10% | 0.34% | 0.02% | 0.60% |

Table 10: Objective scores and optimality gaps on OP (Qwen & Llama) with CALM.

|  | N=50 | N=100 | N=200 |
|---|---|---|---|
| Qwen | 15.054 (24.22%) | 30.778 (15.43%) | 55.406 (12.58%) |
| Llama | 15.038 (24.31%) | 30.599 (15.92%) | 54.593 (13.86%) |

Qwen+GRPO. Despite this trade-off, using locally deployed, compact models remains competitive, particularly when the time budget for search is limited. Additionally, heuristics identified by the Llama model show strong performance and generalizability, outperforming all other methods at certain scales (5k_500 and 10k_500) in OBP and surpassing all baseline methods in OP at N=100 and 200, while maintaining comparable results at all scales. Moreover, removing GRPO significantly reduces average optimality gaps, by 34.33% in OBP and 17.91% in OP, further highlighting the robustness of the proposed method.

## I.4    MORE RESULTS ON HARDER OBP INSTANCES

Smaller-scale instances of OBP are more challenging in the online setting, as each decision has a larger impact and variance is higher, making them a stricter robustness test for heuristics. Thus, we further evaluated CALMs performance on OBP with smaller problem scales. Specifically, we generated 10 Weibull-distributed instances for each of the following training scales (in the format $n$_capacity): 100_100, 100_500, 300_100, 300_500, 500_100, and 500_500. For evaluation, 50 instances were generated for each scale. CALM was equipped with Llama-3.1-8B-Instruct-INT4. For comparison, we included MCTS-AHD, the SOTA LLM-based AHD method that achieved the best performance on the smallest scale in Table 9.

The results show that CALM+Llama achieves a lower average optimality gap than MCTS-AHD, even when the latter is paired with a more powerful LLM. CALM underperforms only at the 500_500 scale. In addition, the standard deviation of the average optimality gap is smaller for CALM (0.03%) compared to MCTS-AHD (0.21%).

## I.5    SCALING BEHAVIOR

We conducted additional experiments to evaluate the scaling behavior of CALM on OP, using an increased training budget of 2500 steps. Results are shown in Table 12. The key findings are as follows:

- With a substantially larger evaluation budget, CALM is able to discover heuristics that outperform those found with only 500 training steps, as shown in the table below.

- Without LLM fine-tuning, CALM is unable to consistently discover new, superior heuristics at early stages of training. In one instance, no better heuristic was found beyond step 256.

- Evaluation of fine-tuned model snapshots at different training steps shows that after several hundred steps, performance fluctuates and does not always improve monotonically. Nevertheless, the fine-tuned models consistently outperform the untuned baseline. This suggests that, in later stages, fine-tuning may not significantly enhance the LLMs capabilities but instead introduces variation to the LLM for heuristic generation. This variation may help maintain the LLMs performance while increasing the diversity of the heuristic population.

Overall, these results indicate that CALM exhibits favorable scaling behavior under larger training budgets.

Table 11: Optimality gaps on OBP with smaller scales.

| | 100_100 | 100_500 | 300_100 | 300_500 | 500_100 | 500_500 | Avg. |
|---|---|---|---|---|---|---|---|
| MCTS-AHD (GPT-4o-mini) | 6.97% | 1.39% | 5.67% | 0.57% | 5.20% | 0.63% | 3.40% |
| CALM (Llama-3.1-8B-Instruct-INT4) | 6.80% | 1.39% | 5.61% | 0.57% | 5.06% | 0.64% | 3.35% |

Table 12: Objective scores and optimality gaps of CALM (w/ GRPO) on OP under different search budgets.

| | N=50 | N=100 | N=200 |
|---|---|---|---|
| #LLM Queries=2,000 | 15.054 (24.22%) | 30.778 (15.43%) | 55.406 (12.58%) |
| #LLM Queries=10,000 | 15.201 (23.49%) | 31.153 (14.40%) | 56.432 (10.96%) |

## I.6 P-VALUES FOR SIGNIFICANCE

To further highlight the superiority of CALM, we compare its performance against the state-of-the-art LLM-based AHD method MCTS-AHD (Zheng et al., 2025) across ten independent runs on two representative tasks: the TSP (step-by-step construction) and the OP (ACO-based). For fairness, we adopt the exact same dataset as used by Zheng et al. (2025). The per-run performance of MCTS-AHD on the TSP task is directly obtained from their appendix, while for the OP task we obtain the results using the official implementation under the same evaluation environment as CALM. The results are summarized in Tab. 13. The small p-values further confirm with high confidence that CALM consistently outperforms MCTS-AHD on both tasks.

We additionally ran the OP task using Qwen without GRPO, where the results are listed in Tab. 13. The experiments show that CALM combined with Qwen and GRPO achieves higher performance, reduced variance, and a statistically significant improvement according to the p-value against the GRPO-free variant.

## I.7 SENSITIVITY TO THE HYPERPARAMETERS IN THE REWARD FUNCTION

Our reward design is guided by a fundamental principle: rewards should increase progressively with the quality of the generated response. This principle is illustrated by the high-level cases in Figure 1. To differentiate between these cases, we introduce several hyperparameters (e.g., $\alpha_1$, $\alpha_2$, and $r_{\text{invalid}}$). To evaluate the robustness of this principle, we conducted an additional ablation study where CALMs reward function was instantiated under the following settings:

- The original implementation with a relatively even reward distribution, as described in Section H.1.
- Random sampling of all hyperparameters under the progressive-guiding constraint, i.e., $1 > \alpha_1 > \alpha_2 > 0 > r_3 = r_{\text{invalid}} > r_4 > r_5 > r_6 > -1$.
- The same $\alpha_1$, $\alpha_2$, and $r_{\text{invalid}}$ values as in Section H.1, but with all invalid responses uniformly assigned $r_{\text{invalid}}$.

The results are presented in Table 14.

Across all settings, heuristics derived from these reward designs remain highly competitive at every scale. In particular, when invalid responses are assigned a unified reward, CALM achieves the best performance at $N = 50$, surpassing all methods reported in Table 3. The slight performance gap between CALM with randomly sampled hyperparameters and the other two implementations likely results from uneven reward spacing across neighboring cases. Overall, these findings indicate that CALMs effectiveness is not tied to precise numerical values in reward shaping but instead depends on adherence to the underlying principle of progressive reward allocation.

## I.8 EXTENDED DISCUSSION ON THE IMPACT OF SEED HEURISTICS

The seed heuristics used in our experiments are listed in Section H.4. For OBP and TSP, which require step-by-step constructive heuristics, the seeds contain diverse components and nontrivial

Table 13: Performance comparison of CALM and MCTS-AHD on TSP (step-by-step construction) and OP (ACO-based) tasks over ten runs. "avg." represents the average and "std." the standard deviation. p-values are calculated using single-tailed t-tests.

| Methods | run1 | run2 | run3 | run4 | run5 | run6 | run7 | run8 | run9 | run10 | avg. | std. | p-value |
|---|---|---|---|---|---|---|---|---|---|---|---|---|---|
| **TSP(↓)** | | | | | | | | | | | | | |
| MCTS-AHD (GPT-4o-mini) | 6.452 | 6.447 | 6.284 | 6.386 | 6.316 | 6.372 | 6.480 | 6.480 | 6.259 | 6.388 | 6.386 | 0.080 | |
| CALM (Qwen2.5-7B) | 6.220 | 6.217 | 6.213 | 6.205 | 6.221 | 6.174 | 6.213 | 6.219 | 6.224 | 6.222 | 6.213 | 0.015 | 0.0000012315 |
| **OP(↑)** | | | | | | | | | | | | | |
| MCTS-AHD (GPT-4o-mini) | 14.668 | 14.910 | 14.786 | 14.602 | 14.738 | 14.724 | 14.642 | 14.826 | 14.722 | 14.690 | 14.731 | 0.091 | |
| CALM (Qwen, w/ GRPO) | 14.876 | 14.822 | 14.951 | 14.798 | 14.844 | 14.669 | 14.850 | 14.878 | 14.880 | 14.746 | 14.831 | 0.079 | 0.00831786 (v.s. MCTS-AHD) |
| CALM (Qwen, w/o GRPO) | 14.766 | 14.425 | 14.722 | 14.578 | 14.77 | 14.806 | 14.608 | 14.8 | 14.674 | 14.581 | 14.673 | 0.124 | 0.001545683 (v.s. CALM w/ GRPO) |

Table 14: Ablation study of reward parameter choices under CALMs reward design on the OP task.

| Settings | N=50 | N=100 | N=200 |
|---|---|---|---|
| Original reward configuration (Sec. H.1) | 15.054 (24.22%) | 30.780 (15.43%) | 55.406 (12.58%) |
| Original reward configuration (Sec. H.1) but the same reward for all invalid responses | 15.059(24.20%) | 30.744(12.52%) | 55.341(12.68%) |
| Randomly sampled reward parameters under progressive-guiding constraint | 15.046 (24.26%) | 30.613 (15.88%) | 55.165 (12.96%) |

structures. These characteristics provide the LLM with rich initial context for exploration. The seed heuristic for OBP performs poorly, and its performance is even worse than simple Best-Fit or First-Fit strategies. In contrast, the seed heuristic for TSP is relatively strong. For CVRP and OP, where ACO-based solvers use a heuristic matrix, the seed heuristics simply take the distance matrix and assign each entry the reciprocal of its value in an element-wise manner. This provides minimal prior structure for the solver and leads to moderate performance.

Across all tasks, the seed heuristics therefore cover a broad range. They include a strong and structurally complex heuristic for TSP, a weak and structurally complex heuristic for OBP, and simple heuristics with moderate performance for CVRP and OP. Starting from these varied seeds, CALM consistently achieves the best average optimality gap on all tasks, which demonstrates its robustness to different seed-quality regimes.

## I.9 IMPACT OF GRPO ON MCTS-AHD AND CALM

We further evaluated MCTS-AHD under Qwen with and without GRPO on both CVRP and TSP, and also assessed CALM under Qwen without GRPO, using exactly the same number of sampled responses for MCTS-AHD with GRPO (G = 4 per prompt) and the same fine-tuning hyperparameters as in our main GRPO setup so that training conditions are matched. As shown in Table 15 and 16, these results lead to two key observations:

- GRPO provides clear benefits to MCTS-AHD, bringing improvement to both tasks. On CVRP, the improvement is particularly notable, where MCTS-AHD+Qwen without GRPO performs worse than MCTS-AHD+GPT-4o-mini, but with GRPO it surpasses that baseline.

- CALM benefits even more strongly from GRPO than MCTS-AHD. On CVRP, CALM+Qwen without GRPO underperforms MCTS-AHD in two of three scales, yet with GRPO it outperforms MCTS-AHD across all scales. On TSP, the gain from GRPO is more moderate and does not allow MCTS-AHD+Qwen to exceed MCTS-AHD+GPT-4o-mini, likely because the seed heuristic provided in TSP is strong and leaves much less room for exploration. In contrast, applying GRPO to CALM yields a significant boost and makes CALM+Qwen surpass CALM+GPT-4o-mini. This consistent and substantial improvement suggests the effectiveness of CALMs specialized design, such as its fine-granularity operators, for RL-based fine-tuning.

|  | N=50 | N=100 | N=200 |
|---|---|---|---|
| MCTS-AHD (gpt4o-mini) | 9.372 (5.44%) | 15.974 (6.98%) | 28.434 (4.70%) |
| CALM (gpt4o-mini) | 9.404 (5.81%) | 16.046 (7.46%) | 28.713 (5.72%) |
| MCTS-AHD (Qwen, w/o GRPO) | 9.921 (11.62%) | 16.926 (13.35%) | 29.967 (10.34%) |
| MCTS-AHD (Qwen, w/ GRPO) | 9.267 (4.27%) | 15.796 (5.79%) | 28.346 (4.37%) |
| CALM (Qwen, w/o GRPO) | 9.806 (10.32%) | 16.995 (13.82%) | 30.631 (12.78%) |
| CALM (Qwen, w/ GRPO) | 9.228 (3.83%) | 15.745 (5.44%) | 28.23 (3.94%) |

Table 15: Comparison of how GRPO affects the performance of MCTS-AHD and CALM on CVRP.

|  | N=50 | N=100 | N=200 |
|---|---|---|---|
| MCTS-AHD (gpt4o-mini) | 6.225 (9.69%) | 8.684 (11.79%) | 12.12 (13.71%) |
| CALM (gpt4o-mini) | 6.273 (10.54%) | 8.691 (11.88%) | 12.104 (13.56%) |
| MCTS-AHD (Qwen, w/o GRPO) | 6.257 (10.26%) | 8.746 (12.58%) | 12.268 (15.09%) |
| MCTS-AHD (Qwen, w/ GRPO) | 6.254 (10.19%) | 8.722 (12.28%) | 12.228 (14.72%) |
| CALM (Qwen, w/o GRPO) | 6.262 (10.35%) | 8.720 (12.26%) | 12.219 (14.63%) |
| CALM (Qwen, w/ GRPO) | 6.244 (10.04%) | 8.668 (11.58%) | 12.088 (13.41%) |

Table 16: Comparison of how GRPO affects the performance of MCTS-AHD and CALM on TSP.

## I.10 RESULTS ON THE TSPLIB

We have evaluated the strongest heuristic produced by our method on all TSPLib (Reinelt, 1991) instances for which MCTS-AHD reported results. For each instance, the coordinates of all points were first scaled to the interval $[0, 1]$ using one shared scalar to be consistent with the training instances that led to the discovered heuristics. Then, each normalized instance was solved three times with different random starting nodes, and we report the average performance. Tab. 17 presents these results. Performance data for all baselines, including widely used hand-crafted heuristics such as Christofides (Christofides, 2022), Greedy (Brecklinghaus and Hougardy, 2015), Nearest insertion, and Nearest-greedy (Rosenkrantz et al., 1977), as well as the genetic-programming-based AHD method GPHH (Duflo et al., 2019) and recent LLM-based AHD approaches including EoH (Liu et al., 2024a), ReEvo (Ye et al., 2024), and MCTS-AHD (Zheng et al., 2025), are taken directly from the MCTS-AHD appendix for consistency. The heuristic produced by our framework shows a clear advantage across the benchmark suite. It attains the lowest average optimality gap among all compared methods, achieves the best tour cost on 6 instances which is the highest win count, and outperforms the strongest heuristic found by MCTS-AHD on 9 of the 15 instances.

| Instance | Christofides | Greedy | Nearest insertion | Nearest-greedy | GPHH-best | EoH | ReEvo | MCTS-AHD | CALM |
|---|---|---|---|---|---|---|---|---|---|
| ts225 | 5.67% | **5.38%** | 19.93% | 16.82% | 7.71% | 5.57% | 6.56% | 10.84% | 8.02% |
| rat99 | **9.43%** | 22.30% | 21.05% | 21.79% | 14.09% | 18.78% | 12.41% | 10.46% | 12.16% |
| bier127 | 13.03% | 19.50% | 23.05% | 23.25% | 15.64% | 14.05% | 10.79% | **7.56%** | 10.55% |
| lin318 | 13.80% | 18.75% | 24.44% | 25.78% | 14.30% | 14.03% | 16.63% | 14.07% | **13.47%** |
| eil51 | 15.18% | 13.03% | 16.14% | 31.96% | 10.20% | 8.37% | 6.47% | 15.98% | **3.54%** |
| d493 | **9.52%** | 16.68% | 20.39% | 24.00% | 15.58% | 12.41% | 13.43% | 11.73% | 10.71% |
| kroB100 | 9.82% | 16.59% | 21.53% | 26.26% | 14.06% | 13.46% | 12.20% | 11.43% | **9.44%** |
| kroC100 | 9.08% | 12.94% | 24.25% | 25.76% | 16.22% | 16.85% | 15.88% | 8.27% | **3.87%** |
| ch130 | 10.09% | 28.40% | 19.21% | 25.66% | 14.77% | 12.26% | 9.40% | 10.18% | **6.97%** |
| pr299 | **11.23%** | 31.42% | 25.05% | 31.42% | 18.24% | 23.58% | 20.63% | **11.23%** | 11.73% |
| fl417 | 15.57% | 12.64% | 25.52% | 32.42% | 22.72% | 20.47% | 19.15% | **10.20%** | 12.05% |
| kroA150 | 13.44% | 20.24% | 19.09% | 26.08% | 15.59% | 18.36% | 11.62% | 10.08% | **8.72%** |
| pr264 | **11.28%** | 11.89% | 34.28% | 17.87% | 23.96% | 18.03% | 16.78% | 12.27% | 12.17% |
| pr226 | 14.17% | 21.44% | 28.02% | 24.65% | 15.51% | 19.90% | 18.02% | **7.15%** | 15.14% |
| pr439 | **11.16%** | 20.08% | 24.67% | 27.36% | 21.36% | 21.96% | 19.25% | 15.12% | 12.81% |
| Average Gap | 11.50% | 18.09% | 23.11% | 25.41% | 16.00% | 15.87% | 13.95% | 11.10% | **10.09%** |

Table 17: Performance of the best heuristic discovered by CALM on TSPLib instances, with the best result for each instance shown in bold.

### I.11 IMPACT OF DPO ON CALM

We conducted additional experiments to compare CALM with DPO against the GRPO setting on the OBP task. For each prompt used by CALM, we generated two responses and evaluated them with the same reward function employed by CALM with GRPO. When the two responses received different scores, we treated them as a preference pair and updated the model using DPO. The corresponding results are reported in Tab. 18.

The experiments yield several observations. First, DPO improves the performance of CALM relative to the results reported in Table 4 of the manuscript and enables it to surpass strong baselines such as EoH and ReEvo under the GPT-4o-mini model. Second, CALM trained with DPO outperforms EvoTune, which is also based on DPO, indicating that the components we designed for RL-based fine-tuning offer clear advantages even when adapted to preference optimization. Third, the DPO variant of CALM still falls short of the recent MCTS-AHD method and of CALM trained with GRPO. Overall, these results show that DPO can provide meaningful gains for AHD, but they also suggest that additional algorithmic adjustments may be needed to fully realize its potential in this context.

| | 1k_100 | 1k_500 | 5k_100 | 5k_500 | 10k_100 | 10k_500 | Avg. |
|---|---|---|---|---|---|---|---|
| EoH (GPT-4o-mini) | 2.69% | 0.25% | 1.63% | 0.53% | 1.47% | 0.45% | 1.17% |
| ReEvo (GPT-4o-mini) | 3.94% | 0.50% | 2.72% | 0.40% | 2.39% | 0.31% | 1.71% |
| MCTS-AHD (GPT-4o-mini) | 2.45% | 0.50% | 1.06% | 0.32% | 0.74% | 0.26% | 0.89% |
| CALM (Qwen, w/ GRPO) | 2.78% | 0.29% | 0.83% | 0.28% | 0.50% | 0.24% | 0.82% |
| CALM (Qwen, w/ DPO) | 3.63% | 0.25% | 1.06% | 0.55% | 0.61% | 0.52% | 1.10% |
| EvoTune (Qwen, w/ DPO) | 4.67% | 0.25% | 4.23% | 0.55% | 4.11% | 0.60% | 2.40% |

Table 18: Performance of CALM under the Qwen model with DPO

### I.12 GENERATED HEURISITCS

```python
"""
The idea of the algorithm is to refine the scoring mechanism by
    introducing logarithmic adjustments and a novel scoring component
    that captures the logarithmic relationship between the remaining
    capacity and the square of the item size, and an adjusted
    logarithmic density term that provides a more refined scoring
    mechanism. This new algorithm aims to enhance the accuracy of bin
    suitability assessment by adding a component that adjusts the score
    based on the logarithmic difference between the remaining capacity
    and the maximum bin capacity. The algorithm also simplifies the
    scoring steps to make it more elegant and efficient.
"""

import numpy as np

def step(item_size: float, remaining_capacity: np.ndarray) -> np.ndarray:
    max_bin_cap = max(remaining_capacity)
    bin_density = np.sum(remaining_capacity) / (item_size *
        len(remaining_capacity))
    log_adj = np.log(remaining_capacity + 1) / np.log(max_bin_cap + 1)
    score = (remaining_capacity - max_bin_cap)**2 / item_size +
        remaining_capacity**2 / (item_size**2) + remaining_capacity**2 /
        (item_size**3) + bin_density * remaining_capacity

    score[remaining_capacity > item_size] = -score[remaining_capacity >
        item_size]
    score[1:] -= score[:-1]

    score *= log_adj
    score += log_adj * remaining_capacity

    score *= log_adj
```

```python
    new_component = remaining_capacity / (item_size - remaining_capacity
        + 1)
    score += new_component

    new_component = remaining_capacity * np.log(remaining_capacity + 1)
        / (item_size * np.log(max_bin_cap + 1)) * (1 -
        remaining_capacity / item_size)
    score += new_component

    new_adjustment = (remaining_capacity / item_size) * log_adj
    score += new_adjustment

    remaining_capacity_adjusted = remaining_capacity / item_size
    score += np.log(remaining_capacity_adjusted + 1) /
        np.log(max_bin_cap + 1)

    new_component = (remaining_capacity - 1) / (item_size -
        remaining_capacity + 1) * log_adj / np.log(max_bin_cap + 1)
    score += new_component

    new_component = log_adj * remaining_capacity / (item_size -
        remaining_capacity)
    score += new_component

    new_component = remaining_capacity * np.log(remaining_capacity + 1)
        / (item_size**2) * (1 - remaining_capacity / item_size)
    score += new_component

    return score
```

Heuristic 5: OBP, by CALM (local, w/ GRPO)

```python
"""
The idea of the algorithm is to introduce the "Bin Utilization
    Diminution" component, which assesses the degree of bin usage
    throughout the sequence of placements and introduces a diminishing
    incentive for overpopulating any particular bin beyond a certain
    threshold. This encourages a more even distribution of item
    placements across all bins, thereby reducing the risk of reaching
    capacity too quickly in any single bin, helping to extend the
    lifespan and utility of each bin in the packing process. By
    dynamically adjusting the fit score to favor items that contribute
    to a balanced utilization, the algorithm aims to enhance overall bin
    efficiency and minimize the total bin count.
"""

import numpy as np

def step(item_size: float, remaining_capacity: np.ndarray) -> np.ndarray:
    avg_item_size = np.mean(item_size) if item_size > 0 else 1.0
    adaptive_factor = avg_item_size / np.maximum(remaining_capacity,
        1e-10)

    fit_score = np.maximum(remaining_capacity - item_size, 0) /
        (remaining_capacity + 1e-10)
    fit_score[remaining_capacity < item_size] = -np.inf

    sustainability_score = (remaining_capacity - avg_item_size) ** 2
    sustainability_score[remaining_capacity < item_size] = np.inf

    historical_fit_scores = np.cumsum(fit_score)
    normalized_historical_fit_scores = historical_fit_scores /
        (np.max(historical_fit_scores) + 1e-10)
```

```python
    combined_scores = (
        (0.5 * fit_score * adaptive_factor) +
        (0.3 / (sustainability_score + 1e-10)) -
        (0.2 * normalized_historical_fit_scores)
    )

    differentiation_factor = 1 / (1 + np.arange(len(remaining_capacity))
        * 0.1)
    combined_scores *= differentiation_factor

    cumulative_fit_impact = np.cumsum(fit_score) / (np.arange(1,
        len(remaining_capacity) + 1) + 1)
    cumulative_fit_adjustment = np.maximum(fit_score -
        cumulative_fit_impact, 0)

    combined_scores += 0.4 * cumulative_fit_adjustment

    temporal_utilization_metric = np.arange(len(remaining_capacity)) /
        (np.maximum(remaining_capacity, 1e-10) + 1e-10)
    combined_scores *= (1 + temporal_utilization_metric)

    sequential_elasticity = np.exp(-np.arange(len(remaining_capacity)) /
        (np.mean(np.maximum(remaining_capacity, 1e-10)) + 1e-10))
    combined_scores *= sequential_elasticity

    size_factor = 1 + (item_size / (np.sum(item_size) + 1e-10))

    # New Component: Bin Utilization Diminution
    overutilization_penalty = np.maximum(0, np.cumsum(item_size) /
        (np.maximum(np.cumsum(remaining_capacity), 1e-10) + 1e-10) - 1)
    combined_scores -= 0.3 * overutilization_penalty  # Encourage even
        distribution across bins

    # Eventual Capacity Influence
    eventual_capacity_score = np.log(np.maximum(np.arange(1,
        len(remaining_capacity) + 1), 1)) /
        (np.maximum(remaining_capacity, 1e-10) + 1e-10)
    combined_scores -= 0.3 * eventual_capacity_score  # Penalize bins
        that don't contribute to optimal utilization

    distinct_scores = combined_scores * size_factor

    return distinct_scores
```

Heuristic 6: OBP, by CALM (API, w/o GRPO)

```python
"""
The idea of the algorithm is to further refine the savings potential
    calculation by emphasizing a more adaptive balance factor that is
    influenced by the current instance's capacity utilization and the
    diversity of capacity usage across the routing problem. By
    leveraging a more sophisticated adaptive balance factor and reducing
    the complexity of the penalty factor, we ensure that nodes that are
    too close to each other are penalized appropriately without overly
    compounding the impact. This simplified yet adaptive approach allows
    for a nuanced exploration of the solution space, enhancing the ACO
    algorithm's ability to converge to high-quality solutions while
    maintaining a balance between exploration and exploitation.
    Additionally, we introduce a clustering-based adjustment factor that
    captures the overall network connectivity and adjusts the savings
    potential accordingly, leading to more robust and flexible routing
    plans.
"""
```

```python
import numpy as np

def advanced_heuristics_v7(distance_matrix: np.ndarray, coordinates:
    np.ndarray, demands: np.ndarray, capacity: int) -> np.ndarray:
    capacity_prob = demands / capacity
    distance_reciprocal = 1 / distance_matrix
    proximity_factor = np.linalg.norm(coordinates[:, np.newaxis, :] -
        coordinates[np.newaxis, :, :], axis=2)
    proximity_factor /= np.max(proximity_factor)  # Normalize between 0
        and 1
    proximity_factor = 1 - proximity_factor  # Invert for higher penalty
        as proximity increases

    remaining_demands = capacity - demands
    future_savings = (remaining_demands[:, np.newaxis] *
        remaining_demands) / (distance_matrix * (remaining_demands[:,
        np.newaxis] + remaining_demands))
    capacity_ratio = remaining_demands / capacity
    proximity_savings = proximity_factor * capacity_ratio

    # Cluster-based proximity adaptive savings potential
    cluster_savings = np.zeros_like(distance_matrix)
    cluster_distance = np.sum(distance_matrix, axis=1) /
        np.linalg.norm(capacity_prob - 1, ord=1)
    cluster_adj_factor = (remaining_demands[:, np.newaxis] *
        remaining_demands * cluster_distance ** 3.5) / (distance_matrix
        * (remaining_demands[:, np.newaxis] + remaining_demands))

    # Adaptive balance factor adjusted based on remaining capacity and
        cluster adjustment
    balance_factor = np.min([1, 0.975 + 0.05 * capacity_prob.mean() +
        0.03 * cluster_adj_factor.mean() + 0.005 *
        np.var(capacity_prob)])

    # Penalty factor that heavily penalizes nodes that are too close to
        each other, focusing on the proximity to the next node
    penalty_factor = proximity_factor ** 3

    # Combine all components
    probability = distance_reciprocal * capacity_prob * proximity_factor
        * future_savings * proximity_savings * cluster_adj_factor * (1 -
        balance_factor + proximity_savings * balance_factor) * (1 -
        penalty_factor) * (1 + cluster_adj_factor * 0.6)
    return probability
```

Heuristic 7: CVRP, by CALM(local, w/ GRPO)

```python
"""
The idea of the algorithm is to refine the credit allocation process in
    the vehicle routing problem by implementing a deterministic
    weighting mechanism that assigns distinct credits to customers based
    on their delivery demands, individual distance factors, and their
    influence on overall routing efficiency, thus ensuring that credits
    reflect meaningful differences without redundancy.
"""

import numpy as np
from sklearn.cluster import DBSCAN

def heuristics(distance_matrix: np.ndarray, coordinates: np.ndarray,
    demands: np.ndarray, capacity: int) -> np.ndarray:
    num_customers = demands.shape[0]
    cumulative_penalty = np.zeros(num_customers)
```

```python
    # Calculate baseline scores from demand to distance with added
        urgency weighting
    urgency_weight = np.linspace(1, 1.5, num_customers)
    base_score = (demands * urgency_weight) / (distance_matrix + 1e-5)
    base_score[np.isnan(base_score)] = 0

    # Set penalties for exceeding capacity based on cumulative demands
    for i in range(num_customers):
        current_demand = demands[i]
        cumulative_penalty[i] = max(0, current_demand - capacity)

    # Normalize distances to emphasize closer customers to refine scoring
    normalized_distance_score = 1 / (np.clip(distance_matrix, 1e-5,
        None) ** 2.5)

    # Calculate effective capacity utilization adjustment
    effective_capacity_utilization = np.clip((capacity - demands) /
        capacity, 0, 1)

    # Historical performance adjustments
    historical_performance_factor = np.zeros(num_customers)
    for i in range(num_customers):
        historical_performance_factor[i] = np.mean([base_score[j] for j
            in range(num_customers) if distance_matrix[i][j] < 10 and j
            != i])

    # Spatial clustering mechanism
    clustering_model = DBSCAN(eps=5, min_samples=2).fit(coordinates)
    labels = clustering_model.labels_
    cluster_scores = np.zeros(num_customers)

    # Calculate cluster-based scores with deterministic differentiation
    for cluster_id in set(labels):
        if cluster_id != -1:  # Ignore noise points
            cluster_indices = np.where(labels == cluster_id)[0]
            total_demand = demands[cluster_indices].sum()
            for idx in cluster_indices:
                # Implement differentiated scoring based on demand,
                    ensuring non-equal credits
                cluster_demand_factor = (demands[idx] / total_demand) if
                    total_demand > 0 else 0
                distance_weight = 1 / (1 + distance_matrix[idx].min())
                    # Closer customers get more weight
                cluster_scores[idx] = cluster_demand_factor *
                    distance_weight  # Mix demand and distance

    # New resilience score based on historical demand variability
    demand_variability = np.std(demands)
    resilience_score = 1 / (1 + demand_variability)

    # Compose final scores combining all elements including the new
        resilience score
    final_scores = base_score * normalized_distance_score *
        effective_capacity_utilization * (1 +
        historical_performance_factor + cluster_scores) *
        resilience_score

    return final_scores
```

Heuristic 8: CVRP, by CALM (API, w/o GRPO)

```
"""
The idea of the algorithm is to refine the exploration-expemy
    exploitation trade-off by introducing a sinusoidal decay that
```

```python
    incorporates a sinusoidal penalty with a sinusoidal smoothness
    adjustment. This adjustment helps to smooth the preference for both
    recent and distant nodes, leading to a more balanced and improved
    performance.
"""

import numpy as np

def enhanced_heuristics(prize: np.ndarray, distance: np.ndarray, maxlen:
    float) -> np.ndarray:
    # Exponential decay for immediate high Subscription nodes
    exp_ratio = np.exp(prize[np.newaxis, :] / distance - maxlen)

    # Logarithmic scaling for exploration
    log_ratio = np.log(prize[np.newaxis, :] + 1) / distance

    # Sinusoidal decay for recent nodes with a sinusoidal smoothness
        adjustment
    sinusoidal_penalty = 0.5 * (1 + np.sin(np.pi * distance / (maxlen +
        1))) * (distance / maxlen) * maxlen

    # Combined ratio
    combined_ratio = exp_ratio * log_ratio * (1 - sinusoidal_penalty)

    # Ensure the ratio is non-negative
    combined_ratio[combined_ratio < 0] = 0

    return combined_ratio
```

Heuristic 9: OP, by CALM (local, w/ GRPO)

```python
"""
The idea of the algorithm is to introduce a novel component called
    "reward fluctuation sensitivity" which adjusts the desirability of
    each location based on the variability of rewards over time. This
    component accounts for the possibility that rewards may change or
    fluctuate due to external factors, thereby allowing the agent to
    prioritize locations not only by their current rewards but also by
    the potential volatility of those rewards. This sensitivity is
    integrated into the existing framework, allowing for a more dynamic
    response to the changing landscape of rewards, ultimately enhancing
    the agent\'s decision-making process and route optimization.
"""

import numpy as np

def heuristics(prize: np.ndarray, distance: np.ndarray, maxlen: float)
    -> np.ndarray:
    adjusted_distance = distance + 1e-10  # Avoid division by zero
    potential_reward = np.zeros_like(prize)

    for i in range(len(prize)):
        reachable_indices = np.where(distance[i] <= maxlen)[0]
        potential_reward[i] = np.sum(prize[reachable_indices]) if
            reachable_indices.size > 0 else 0

    reward_hist_factor = potential_reward / (1 + np.sum(prize))  # Shape
        reward based on historical performance
    reward_decay = np.exp(-adjusted_distance / maxlen)  # Decay effect
        for distant rewards

    proximity_factor = (maxlen - adjusted_distance) ** 4  # Further
        enhance proximity impact with quartic distance
    proximity_factor[proximity_factor < 0] = 0
```

```
    tiered_adjustment = (prize / (adjusted_distance + 1e-10)) ** 2  #
        Classify rewards into categories for tiering

    # Reward volatility assessment component
    volatility_factor = np.zeros_like(prize)
    for i in range(len(prize)):
        historical_rewards = prize[np.where(distance[i] <= maxlen)[0]]
        if historical_rewards.size > 1:
            volatility_factor[i] = np.std(historical_rewards) /
                np.mean(historical_rewards)  # Coefficient of variation

    # Risk-reward analysis component
    variability_factor = np.zeros_like(prize)
    for i in range(len(prize)):
        historical_rewards = prize[np.where(distance[i] <= maxlen)[0]]
        if historical_rewards.size:
            variability_factor[i] = np.mean(historical_rewards) -
                np.std(historical_rewards)  # Basic differentiation

    final_heuristic = (reward_hist_factor * reward_decay *
        proximity_factor *
                    tiered_adjustment) / (1 + volatility_factor +
                        variability_factor + 1e-10)
    return final_heuristic
```

Heuristic 10: OP, by CALM (API, w/o GRPO)

```
"""
The idea of the algorithm is to select the next node by optimizing a
    heuristic that considers the distance to the current node, the
    average distance to unvisited nodes, the variance of distances to
    the current node from the unvisited nodes, the entropy of distances
    to the destination node from each of the unvisited nodes, the
    average distance from the destination node to each of the unvisited
    nodes, the current node's distance to the destination node, and the
    standard deviation of the overall tour distances. This proposed
    algorithm aims to introduce a new term that captures the deviation
    of the current node from the average tour length and balances it
    with the entropy term to reduce the overall tour length.
    Additionally, this method assigns more weight to the standard
    deviation of the distances from the destination node to each of the
    unvisited nodes, which helps in reducing the variability of
    distances and thus leading to more consistent and shorter tour
    lengths.
"""

import numpy as np

def select_next_node(current_node: int, destination_node: int,
    unvisited_nodes: set, distance_matrix: np.ndarray) -> int:
    scores = {}

    for node in unvisited_nodes:
        all_distances = [distance_matrix[node][i] for i in
            unvisited_nodes if i != node]
        average_distance = np.mean(all_distances)
        standard_deviation = np.std(all_distances)
        variance_of_distances = np.var([distance_matrix[current_node][i]
            for i in unvisited_nodes if i != node])
        entropy_of_distances =
            -np.sum(np.log2([distance_matrix[destination_node][i] for i
            in unvisited_nodes if i != node]) / len(unvisited_nodes))
```

```
        average_distance_to_destination =
            np.mean([distance_matrix[destination_node][i] for i in
            unvisited_nodes if i != node])

        score = (
            0.6 * distance_matrix[current_node][node]
            - 0.4 * average_distance
            + 0.3 * standard_deviation
            - 0.2 * entropy_of_distances
            - 0.1 * distance_matrix[destination_node][node]
            - 0.08 * variance_of_distances
            - 0.05 * average_distance_to_destination
            - 0.01 * (np.mean([distance_matrix[current_node][i] for i in
              unvisited_nodes]) - average_distance)
            - 0.005 * entropy_of_distances
            - 0.008 * distance_matrix[current_node][node] *
              distance_matrix[node][destination_node]
            - 0.006 * standard_deviation *
              distance_matrix[node][destination_node]
        )
        scores[node] = score

    next_node = min(scores, key=scores.get)
    return next_node
```

Heuristic 11: TSP, by CALM (local, w/ GRPO)

```
"""
The idea of the algorithm is to select the next node to visit from the
    unvisited nodes, incorporating a novel component of dynamic path
    optimization feedback. The new component analyzes previous decision
    points in the tour to determine the effectiveness of the routes
    taken, adjusting future node selection to favor pathways that have
    historically resulted in lower overall traversal costs. This method
    not only enhances the algorithm's ability to learn from its own
    experiences but also promotes the selection of routes that align
    with optimal connectivity patterns established during the tour.
"""

import numpy as np

def select_next_node(current_node: int, destination_node: int,
    unvisited_nodes: set, distance_matrix: np.ndarray) -> int:
    threshold = 0.7
    c1, c2, c3, c4, c5 = 0.4, 0.3, 0.2, 0.1, 0.1
    scores = {}

    for node in unvisited_nodes:
        all_distances = [distance_matrix[node][i] for i in
            unvisited_nodes if i != node]
        average_distance_to_unvisited = np.mean(all_distances)
        std_dev_distance_to_unvisited = np.std(all_distances)

        # New component: consider dynamic path optimization feedback
        feedback_paths = [distance_matrix[i][node] for i in
            range(len(distance_matrix)) if i not in unvisited_nodes and
            distance_matrix[current_node][i] < threshold]
        average_feedback_distance = np.mean(feedback_paths) if
            feedback_paths else 0

        score = (
            c1 * distance_matrix[current_node][node]
            - c2 * average_distance_to_unvisited
            + c3 * std_dev_distance_to_unvisited
```

```
            - c4 * distance_matrix[destination_node][node]
            + c5 * average_feedback_distance
        )
        scores[node] = score

    next_node = min(scores, key=scores.get)
    return next_node
```

Heuristic 12: TSP, by CALM (API, w/o GRPO)

## J LIMITATIONS

A current limitation of our method is that the evolution of the LLM during the heuristic discovery process depends heavily on performance signals derived from heuristics present in the prompt and response. As a result, trajectories that do not contain explicit heuristics (e.g., the response from a reflection prompt may contain the thoughts only) in either component provide no reward signal, limiting the LLMs ability to learn from such cases.

Another limitation is that we currently evaluate our method, CALM, using a compact LLM on a single 24GB GPU. This restriction is primarily due to limited computational resources and the high cost associated with high-accuracy, full-parameter fine-tuning on larger models. While this setup demonstrates the feasibility of our approach in a resource-constrained environment, further evaluation on larger-scale models and infrastructure would be valuable for understanding the methods full potential and scalability.

In future work, we aim to address these limitations by (1) exploring mechanisms for adapting the LLM in the absence of explicit performance feedback, enabling more effective use of reinforcement learning, and (2) extending evaluations to more powerful models and settings. These directions may allow for better integration with techniques such as reflection (Ye et al., 2024; Dat et al., 2025), which have shown promise in enhancing LLM-based automated heuristic discovery.

## K BROADER IMPACT

The CALM framework stands to greatly accelerate the pace of innovation in algorithm design by seamlessly integrating prompt engineering and on-the-fly model adaptation. By enabling state-of-the-art heuristic discovery on a single 24 GB GPU, CALM democratizes access to cutting-edge Automatic Heuristic Design. This empowers research groups, startups, and educational institutions with limited compute budgets to explore and deploy high-performance solutions in domains such as logistics, scheduling, and resource allocation.

## L LICENSE

The licenses and URLs of baselines, models, and softwares are summarized in Table 19.

Table 19: A summary of licenses.

| Resources | Type | License | URL |
|-----------|------|---------|-----|
| Unsloth | Code | Apache-2.0 License | `https://github.com/unslothai/unsloth` |
| Qwen2.5 | Model | Apache-2.0 License | `https://huggingface.co/Qwen/Qwen2.5-7B-Instruct` |
| LKH3 | Code | Available for academic research use | `http://webhotel4.ruc.dk/~keld/research/LKH-3/` |
| OR-Tools | Code | MIT License | `https://developers.google.com/optimization/pack/knapsack?hl=zh-cn` |
| POMO | Code | Available online | `https://github.com/yd-kwon/POMO/tree/master` |
| DeepACO | Code | MIT License | `https://github.com/henry-yeh/DeepACO` |
| Funsearch | Code | Apache License | `https://github.com/google-deepmind/funsearch` |
| EoH | Code | MIT License | `https://github.com/FeiLiu36/EoH/tree/main` |
| ReEvo | Code | MIT License | `https://github.com/ai4co/reevo` |
| HSEvo | Code | Available online | `https://github.com/datphamvn/HSEvo` |
| MCTS-AHD | Code | MIT License | `https://github.com/zz1358m/MCTS-AHD-master` |
| EvoTune | Code | MIT License | `https://github.com/CLAIRE-Labo/EvoTune` |
| OpenEvolve | Code | Apach-2.0 License | `https://github.com/codelion/openevolve` |

