# OpenReview forum: "CALM: Co-evolution of Algorithms and Language Model for Automatic Heuristic Design"
_ICLR.cc/2026/Conference — ICLR 2026 Poster_

### Official Review · Reviewer_SGF5 · 2025-10-20

**Soundness:** 3
**Presentation:** 3
**Contribution:** 3
**Rating:** 8
**Confidence:** 3

**Summary:**

This manuscript proposed CALM to generate heuristics.

**Strengths:**

**S1:**  This manuscript is well-structured.

**S2:**  This paper fine-tunes LLMs to generate heuristics, which represents a novel contribution.

**S3:**  The experiments demonstrate the effectiveness of each proposed method.

**Weaknesses:**

**W1:**  This manuscript should introduce more LLM-based Heuristics Generation methods, such as [1,2].

**W2:**  This manuscript lacks experimental comparisons between GRPO and fine-tuning strategies such as DPO.

[1] Efficient heuristics generation for solving combinatorial optimization problems using large language models. KDD, 2025.

[2] Multi-objective Evolution of Heuristic Using Large Language Model. AAAI, 2025.

**Questions:**

**Q1:** The authors measure diversity by counting unique tokens in a heuristic’s idea. Would using a deep neural network to embed each heuristic idea and compute diversity in that embedding space be more effective?

---

> ### Author Response · Authors · 2025-11-20
> **Response (1/2)**
>
> ## Response to Weakness 1
>
> Thanks for your helpful suggestion. In the revised manuscript, we have expanded Section 2 to include more LLM-based heuristic generation approaches. For your convenience, the updated content is as follows:
>
> > "Additionally, some recent studies have also explored reduction techniques [1], trajectory-based analysis [2], and multi-objective optimization [3], to further enhance AHD. [4] have examined how to abstract core components from elite heuristics and combine them with LLM-based fitness prediction for AHD. However, prior work keeps the LLM static. Our approach improves this by continuously fine-tuning the LLM using prompt-response-performance tuples from the evolutionary process, enhancing future heuristic generation."
>
> The references are:
>
> [1] Thach, N., Riahifar, A., Huynh, N., \& Chan, H. (2025). RedAHD: Reduction-Based End-to-End Automatic Heuristic Design with Large Language Models. arXiv preprint arXiv:2505.20242.
>
> [2] Yang, X., Zhang, L., Qian, H., Song, L., \& Bian, J. (2025). HeurAgenix: Leveraging LLMs for Solving Complex Combinatorial Optimization Challenges. arXiv preprint arXiv:2506.15196.
>
> [3] Yao, S., Liu, F., Lin, X., Lu, Z., Wang, Z., \& Zhang, Q. (2025, April). Multi-objective evolution of heuristic using large language model. In Proceedings of the AAAI Conference on Artificial Intelligence (Vol. 39, No. 25, pp. 27144-27152).
>
> [4] Wu, X., Wang, D., Wu, C., Wen, L., Miao, C., Xiao, Y., \& Zhou, Y. (2025, August). Efficient heuristics generation for solving combinatorial optimization problems using large language models. In Proceedings of the 31st ACM SIGKDD Conference on Knowledge Discovery and Data Mining V. 2 (pp. 3228-3239).
>
> ## Response to Weakness 2
>
> Thank you for this helpful suggestion. We conducted additional experiments to compare CALM with DPO against the GRPO setting on the OBP task. For each prompt used by CALM, we generated two responses and evaluated them with the same reward function employed by CALM with GRPO. When the two responses received different scores, we treated them as a preference pair and updated the model using DPO. The corresponding results are reported in the table below.
>
> The experiments yield several observations. First, DPO improves the performance of CALM relative to the results reported in Table 4 of the manuscript and enables it to surpass strong baselines such as EoH and ReEvo under the GPT-4o-mini model. Second, CALM trained with DPO outperforms EvoTune, which is also based on DPO, indicating that the components we designed for RL-based fine-tuning offer clear advantages even when adapted to preference optimization. Third, the DPO variant of CALM still falls short of the recent MCTS-AHD method and of CALM trained with GRPO. Overall, these results show that DPO can provide meaningful gains for AHD, but they also suggest that additional algorithmic adjustments may be needed to fully realize its potential in this context.
>
> |                        | 1k\_100 | 1k\_500 | 5k\_100 | 5k\_500 | 10k\_100 | 10k\_500 | Avg.   |
> |------------------------|---------|---------|---------|---------|----------|----------|--------|
> | EoH (GPT-4o-mini)      | 2.69\%  | 0.25\%  | 1.63\%  | 0.53\%  | 1.47\%   | 0.45\%   | 1.17\% |
> | ReEvo (GPT-4o-mini)    | 3.94\%  | 0.50\%  | 2.72\%  | 0.40\%  | 2.39\%   | 0.31\%   | 1.71\% |
> | MCTS-AHD (GPT-4o-mini) | 2.45\%  | 0.50\%  | 1.06\%  | 0.32\%  | 0.74\%   | 0.26\%   | 0.89\% |
> | CALM (Qwen, w/ GRPO)   | 2.78\%  | 0.29\%  | 0.83\%  | 0.28\%  | 0.50\%   | 0.24\%   | 0.82\% |
> | CALM (Qwen, w/ DPO)    | 3.63\%  | 0.25\%  | 1.06\%  | 0.55\%  | 0.61\%   | 0.52\%   | 1.10\% |
> | EvoTune (Qwen, w/ DPO) | 4.67\%  | 0.25\%  | 4.23\%  | 0.55\%  | 4.11\%   | 0.60\%   | 2.40\% |

---

> > ### Author Response · Authors · 2025-11-20
> > **Response (2/2)**
> >
> > ## Response to Question 1
> >
> > This is an interesting question. We would like to explain why such an approach might not be well-suited in our setting. While embedding models may appear more semantically meaningful than token‐level overlap, they can behave unreliably for heuristic–generation tasks where small textual edits lead to disproportionately large behavioral shifts. For instance, modifying a single symbol such as changing "$<$" to "$>$" can fundamentally alter the logic of a heuristic and produce dramatically different performance, even though the edit distance is minimal. Recent work [1] on LLM-based search reports that embedding models struggle in such domains: although embeddings were used to project candidate solutions into a latent space for Bayesian optimization, the method consistently failed for code‐search tasks, with empirical evidence indicating that current embeddings do not provide "distinct representations for sequences with low-edit distances". These limitations make embedding-based similarity an unreliable proxy for diversity measurement in our problem.
> >
> > Overall, identifying more suitable representations for guiding the search remains an important direction, and we plan to explore it in future work.
> >
> >
> > [1] Agarwal D, Arivazhagan M G, Das R, et al. Searching for optimal solutions with LLMs via bayesian optimization[C]//The Thirteenth International Conference on Learning Representations. 2025.

---

> > > ### Comment · Reviewer_SGF5 · 2025-11-25
> > >
> > > Thank you for the clarification. I will keep my score.

---

### Official Review · Reviewer_ny5L · 2025-10-29

**Soundness:** 2
**Presentation:** 1
**Contribution:** 2
**Rating:** 2
**Confidence:** 4

**Summary:**

This paper proposes a GRPO-based automatic heuristic design framework which employs LLMs to iteratively modify optimization codes to generate better performing algorithms. Authors decompose mutation operators into injection and replacement. Diversity-aware crossover and collapse mechanism is introduced to enhance diversity, and a progressive reward function is designed to measure the code quality. Experimental results show that the proposed method surpasses heuristic design baselines on multiple optimization tasks.

**Strengths:**

1. CALM is presented as one of the first LLM-based AHD frameworks to incorporate GRPO

2. This work proposes new heuristic design operators such as injection, replacement and collapse, which could provide insights for future works. The proof and discussion about the introduction of collapse is provided.

3. Experimental results show that CALM outperforms LLM-based heuristic design baselines on multiple combinatorial optimization tasks.

**Weaknesses:**

1. In line 206-207, the authors claim they use compact summaries instead of full code; however, the response examples in Appendix E show generated operators as exact code, like mutation operators in prior LLM-based AHD methods. Additionally, focusing on granular modifications might limit the search space and reduce code diversity.

2. The proposed method uses seed heuristics to generate initial heuristics; however, the authors do not specify the methodology for selecting these seed heuristics. The specific seed heuristics used in experiments are not reported, and the impact of seed heuristics is not discussed. Since the subsequent heuristics are all derived from the seed heuristics, this selection could significantly affect the performance.

3. In reward design, $r_{invalid} \in (-1, 0)$, $\alpha_1, \alpha_2, \Delta(\cdot) \in (0, 1)$ and $\alpha_1 > \alpha_2$, meaning $\alpha_1 r_{invalid}$ (the first term in Eq. (4) ) is more negative than $\alpha_2 r_{invalid} \Delta(\cdot)$ (the second term). This preference for worse steps over no-improvement steps appears questionable. Additionally, in Appendix H.1, authors propose 5 new reward parameters, $r_3 \sim r_7$ (where are $r_1$ and $r_2$ ? ), which do not appear in Eq. (4). The detail of how they are used is absent. Furthermore, the values of $\alpha_1$ and $\alpha_2$ are not reported.

4. In the experiments, baselines are compared under different base LLMs with different knowledge and capabilities, potentially introducing bias. The source of CALM's advantage remains unclear—would Qwen-based baselines outperform it?  For instance, the performance of o4-mini with CALM without GRPO in Table 9 dominates all baselines in both Table 1 and Table 9, potentially reduces the contribution of this paper.

5. The presentation should be improved. For example, line 312 and 319 miss periods at the end of the sentences. Algorithm 1 abuses the equal sign which refers to both equations and assignments.

**Questions:**

See Weaknesses.

---

> ### Author Response · Authors · 2025-11-20
> **Response (1/4)**
>
> ## Response to Weakness 1
>
> (1) Regarding the claim.
>
> Thank you for pointing out the ambiguity. Our original wording did not accurately express our intention. We did not mean that our method relies *solely* on compact summaries or that code is never provided. Our goal was to emphasize the difference in how CALM presents information to the LLM when exploring new algorithms. Prior AHD approaches typically supply the complete code of several existing algorithms when the LLM is asked to design a novel heuristic, whereas CALM offers a single base algorithm together with concise summaries of explored components.
>
> In existing frameworks such as EoH and MCTS-AHD, the common practice for encouraging the LLM to generate previously unseen heuristics is to present a collection of full algorithm implementations and instruct the model to create something structurally different. For instance, EoH uses a prompt (E1) of the form:
> > "I have k existing algorithms with their codes as follows... Please help me create a new algorithm that has a totally different form from the given ones but can be motivated from them."
>
> MCTS-AHD adopts a similar prompt (Action e1):
> >"I have k existing algorithms with their codes as follows: ... Please create a new algorithm that has a totally different form from the given algorithms. Try generating codes with different structures, flows or algorithms."
>
> By contrast, when CALM performs operator injection, it supplies the complete code of only one base heuristic and then augments it with compact summaries describing the components that have already been explored. This setting still allows the LLM to generate operator code, but the exploration is guided by summaries rather than by multiple full reference implementations.
>
> In summary, while EoH and MCTS-AHD rely on the full code of several existing heuristics to enable exploration, CALM uses the code of one heuristic together with multiple compact descriptions of the explored components. To avoid misunderstanding, we have revised the claim in lines 206–207 accordingly and corrected the typo in Figure 4 (from "given\_heuristics" to "given\_heuristic"). The revised claim is quoted as follows:
> > "Compared to prior LLM-based AHD methods: (1) To explore under-explored heuristic designs, CALM provides the code of only one base heuristic together with compact summaries of components that have already been explored, whereas prior methods typically supply the full code of multiple existing heuristics to prompt the LLM to produce a different one. This design allows more references to be accommodated within the LLM's context window."
>
> (2) Regarding the granular modifications.
>
> We respectfully disagree that granular modifications limit the search space or reduce the diversity of generated code. In fact, heuristic ideas in our setting exhibit high behavioral sensitivity to small textual edits. A seemingly minor change, such as replacing the symbol “$<$” with “$>$”, can invert the underlying logic and lead to substantially different algorithmic behavior, even though the modification is minimal at the word level. This property implies that fine-grained edits do not collapse the space of possibilities; rather, they allow the search process to explore a broad range of functionally distinct heuristics through concise adjustments.
>
> Empirical evidence from an ICLR25 paper [1] on LLM-based search further supports this view. These works find that sequences with low textual edit distance can correspond to meaningfully different behaviors, and that current embedding models often fail to capture these distinctions. Such findings suggest that behavioral diversity need not correlate with the magnitude of textual change. Consequently, granular modifications do not constrain diversity but instead serve as an efficient mechanism for traversing the underlying landscape of heuristic behaviors.
>
>
> [1] Agarwal D, Arivazhagan M G, Das R, et al. Searching for optimal solutions with LLMs via bayesian optimization[C]//The Thirteenth International Conference on Learning Representations. 2025.

---

> ### Author Response · Authors · 2025-11-20
> **Response (2/4)**
>
> ## Response to Weakness 2
>
> All seed heuristics used in our experiments were directly adopted from MCTS-AHD. These heuristics were originally introduced by ReEvo (NeurIPS 2014) and subsequently employed by HSEvo (AAAI 2025) and MCTS-AHD (ICML 2025), and have become a popular experimental setting in recent LLM-based AHD research. We have (1) added a new subsection to the appendix to list all used seed heuristics and (2) revised the manuscript to explicitly describe how these seed heuristics are selected and to clarify that our work follows this established convention. For your convenience, the updated content is as follows:
> >"To ensure a fair comparison, we align CALM and all LLM-based AHD baselines with consistent settings, including shared seed heuristics that are directly adopted from Zheng et al. (2025), ..."
>
> Furthermore, in response to your comment, we have added a subsection to the appendix to discuss the impact of seed heuristics. The content is as follows:
>
> > "The seed heuristics used in our experiments are listed in Appendix H.4. For OBP and TSP, which require step-by-step constructive heuristics, the seeds contain diverse components and nontrivial structures. These characteristics provide the LLM with rich initial context for exploration. The seed heuristic for OBP performs poorly, and its performance is even worse than simple Best-Fit or First-Fit strategies. In contrast, the seed heuristic for TSP is relatively strong. For CVRP and OP, where ACO-based solvers use a heuristic matrix, the seed heuristics simply take the distance matrix and assign each entry the reciprocal of its value in an element-wise manner. This provides minimal prior structure for the solver and leads to moderate performance.
>
> > Across all tasks, the seed heuristics therefore cover a broad range. They include a strong and structurally complex heuristic for TSP, a weak and structurally complex heuristic for OBP, and simple heuristics with moderate performance for CVRP and OP. Starting from these varied seeds, CALM consistently achieves the best average optimality gap on all tasks, which demonstrates its robustness to different seed-quality regimes."
>
> ## Response to Weakness 3
>
> (1) Regarding the reward design.
>
> The intention behind the first term in Eq. (4) is to discourage simple duplication of heuristics already present in the prompt. As described in Eq. (4), this term applies specifically when the generated heuristic achieves exactly the same performance as one of the provided heuristics, a situation that strongly suggests replication rather than genuine exploration. Because improvements are inherently difficult to obtain in a single step, the reward associated with performance gains is naturally sparse. If equal or higher reward were given to exact duplication than to attempts that lead to temporarily worse performance, the model would quickly discover that reproducing a prompt heuristic is the most reliable way to maximize reward. This would undermine the evolutionary process by preventing the creation of novel heuristics. The relative weighting in Eq. (4) is therefore chosen to encourage exploration, even when exploration leads to imperfect intermediate steps.
>
> (2) Regarding the reward parameters
>
> Appendix H.1 provides the details of how these parameters are applied. The additional parameters $r_3$ through $r_7$ are used to differentiate types of invalid responses and are not included in Eq. (4) because that equation specifies only the reward structure for valid responses. We do not define $r_1$ and $r_2$ because our goal is to maintain a simple and memorable ordering among all parameters, expressed as $\alpha_1 > \alpha_2 > r_3 > \cdots > r_7$ as shown in Appendix I.7. The values of $\alpha_1$ and $\alpha_2$ were omitted in the original submission, and we appreciate the reviewer pointing this out. They are now explicitly stated in Appendix H.1 as $0.8$ and $0.5$. We also note that our choice of how to reward invalid responses is one reasonable option rather than a unique solution, and the ablation studies compare this design with alternatives such as using a uniform reward for all invalid responses. The results shown in Appendix I.7 indicate that a range of strategies is effective as long as they follow the same progressive guiding principles.

---

> > ### Author Response · Authors · 2025-11-20
> > **Response (3/4)**
> >
> > ## Response to Weakness 4
> >
> > (1) We would like to clarify that our comparisons do not place our method at an unfair advantage. We genuinely value the reviewer’s concern about the varying knowledge and abilities of LLMs, and we emphasize that the model we employ (Qwen2.5-7B-Instruct-INT4) is substantially weaker than GPT-4o-mini according to the Qwen2.5 technical report, which reports the approximate ordering GPT-4o-mini $\approx$ Qwen2.5-Turbo $>$ Qwen2.5-14B-Instruct $>$ Qwen2.5-7B-Instruct $>$ Qwen2.5-7B-Instruct-INT4. This means that our approach is evaluated under a less capable model, ensuring that improvements cannot be attributed to a stronger underlying LLM.
> >
> > (2) In response to your concern, we conducted additional experiments to provide fairer comparisons within the same Qwen model family. Specifically, we further evaluated MCTS-AHD under Qwen with and without GRPO on both CVRP and TSP, and also assessed CALM under Qwen without GRPO, using exactly the same number of sampled responses for MCTS-AHD with GRPO (G = 4 per prompt), the same reward function, and the same fine-tuning hyperparameters as in our main GRPO setup so that training conditions are matched. These results lead to two key observations: (i) GRPO provides clear benefits to MCTS-AHD, bringing improvement to both tasks. On CVRP, the improvement is particularly notable, where MCTS-AHD+Qwen without GRPO performs worse than MCTS-AHD+GPT-4o-mini, but with GRPO it surpasses that baseline. (ii) CALM outperforms MCTS-AHD under the same Qwen model with GRPO and benefits even more strongly from GRPO than MCTS-AHD. On CVRP, CALM+Qwen without GRPO underperforms MCTS-AHD in two of three scales, yet with GRPO it outperforms MCTS-AHD across all scales. On TSP, the gain from GRPO is more moderate and does not allow MCTS-AHD+Qwen to exceed MCTS-AHD+GPT-4o-mini, likely because the seed heuristic provided in TSP is strong and leaves much less room for exploration. In contrast, applying GRPO to CALM yields a significant boost and makes CALM+Qwen surpass CALM+GPT-4o-mini. This consistent and substantial improvement suggests the effectiveness of CALM’s specialized design, such as its fine-grained operators, for RL-based fine-tuning.
> >
> > **More results on the CVRP task**
> >
> > | Method                    | N=50            | N=100            | N=200            |
> > |---------------------------|-----------------|------------------|------------------|
> > | MCTS-AHD (GPT-4o-mini)    | 9.372 (5.44%)  | 15.974 (6.98%)  | 28.434 (4.70%)  |
> > | CALM (GPT-4o-mini)        | 9.404 (5.81%)  | 16.046 (7.46%)  | 28.713 (5.72%)  |
> > | MCTS-AHD (Qwen, w/o GRPO) | 9.921 (11.62%) | 16.926 (13.35%) | 29.967 (10.34%) |
> > | MCTS-AHD (Qwen, w/ GRPO)  | 9.267 (4.27%)  | 15.796 (5.79%)  | 28.346 (4.37%)  |
> > | CALM (Qwen, w/o GRPO)     | 9.806 (10.32%) | 16.995 (13.82%) | 30.631 (12.78%) |
> > | CALM (Qwen, w/ GRPO)      | 9.228 (3.83%)  | 15.745 (5.44%)  | 28.23 (3.94%)   |
> >
> > **More results on the TSP task**
> > | Method                    | N=50            | N=100           | N=200            |
> > |---------------------------|-----------------|-----------------|------------------|
> > | MCTS-AHD (GPT-4o-mini)    | 6.225 (9.69%)  | 8.684 (11.79%) | 12.12 (13.71%)  |
> > | CALM (GPT-4o-mini)        | 6.273 (10.54%) | 8.691 (11.88%) | 12.104 (13.56%) |
> > | MCTS-AHD (Qwen, w/o GRPO) | 6.257 (10.26%) | 8.746 (12.58%) | 12.268 (15.09%) |
> > | MCTS-AHD (Qwen, w/ GRPO)  | 6.254 (10.19%) | 8.722 (12.28%) | 12.228 (14.72%) |
> > | CALM (Qwen, w/o GRPO)     | 6.262 (10.35%) | 8.720 (12.26%) | 12.219 (14.63%) |
> > | CALM (Qwen, w/ GRPO)      | 6.244 (10.04%) | 8.668 (11.58%) | 12.088 (13.41%) |
> >
> > (3) We respectfully disagree that the strong performance of CALM on o4-mini without GRPO diminishes the contribution of our work. (i) The aim of this paper is to advance automatic heuristic design through RL-based fine-tuning, and the method is not tied to a particular foundation model. Although most experiments use a local Qwen model, this choice reflects practical considerations rather than a methodological constraint. Appendix I.3 further shows that CALM also achieves solid results with a Llama model, indicating that its effectiveness is not limited to a specific model family. (ii) Using a more capable base model also does not conflict with the intended role of CALM, since the method can be applied to any model for which weights are available and adequate computation exists. (iii) In addition, powerful commercial models are often accessible only through APIs for most users. For individuals or organizations seeking to design effective heuristics for proprietary tasks while maintaining data privacy, a fully local LLM-based AHD solution is essential. CALM offers an efficient, low-cost option in this setting.

---

> > > ### Author Response · Authors · 2025-11-20
> > > **Response (4/4)**
> > >
> > > ## Response to Weakness 5
> > >
> > > Thank you for the helpful comments. We have added the missing periods, revised Algorithm 1 so that value assignments consistently use the leftarrow symbol and the equal sign appears only for comparisons, and carefully checked the manuscript for additional typographical issues to improve the overall presentation.

---

> > > > ### Comment · Reviewer_ny5L · 2025-11-25
> > > >
> > > > Thanks the authors for the detailed response, which addresses my concerns. The revised content would make the paper more clear and the reward design seems reasonable. The additional experimental results enhance the soundness of the proposed paper. As a result, I raise the score to 4.

---

### Official Review · Reviewer_9Hpb · 2025-10-31

**Soundness:** 3
**Presentation:** 4
**Contribution:** 3
**Rating:** 8
**Confidence:** 3

**Summary:**

The paper proposes CALM, an LLM-based Automatic Heuristic Design (AHD) framework that co-evolves the prompted search over heuristics with on-the-fly RL fine-tuning of the LLM via GRPO. Beyond standard “verbal gradients” (prompt-driven operators), CALM adds “numerical gradients” by rewarding generated heuristics based on feasibility/novelty and relative performance to parent heuristics. It further introduces fine-granularity mutation (injection/replacement), a diversity-aware crossover, and a probabilistic collapse mechanism to escape stagnation. Experiments show consistent gains over prior LLM-AHD baselines, often using only a quantized 7B model on a single 24 GB GPU; ablations indicate GRPO is the main contributor.

**Strengths:**

1. The paper is easy to follow and well-written

2. The idea is good and has practical relevance

3. Emphasis on local models with low compute budget

4. Code is provided

5. Good results compared to some SOTA algorithms

**Weaknesses:**

1. Authors do not report performance on public benchmarkse.g. TSPLib, which would make it easier to compare CALM against other methods

2. The performance gain in non-GRPO settings is not too clear. For example, it seems that MCTS-AHD outperforms CALM without GRPO, which makes me wonder about why not finetune MCTS-AHD ‘s LLM too, and if s,o what the performance would be

3. The novelty of finetuning LLMs for AHD is not too much due to the existence of concurrent works in finetuning, although CALM uses a different algorithm (GRPS vs DPO)

**Questions:**

1. Why have hardcoded operators via sampling? Would using an LLM generate modifications instead be sufficient?

2. Convergence with fewer LLM queries appears to be worse than other methods (see Fig 2). How could you explain this?

---

> ### Author Response · Authors · 2025-11-20
> **Response (1/3)**
>
> ## Response to Weakness 1
>
> Thank you for this helpful suggestion. We have evaluated the strongest heuristic produced by our method on all TSPLib instances for which MCTS-AHD reported results. Following the MCTS-AHD protocol, each instance was solved three times with different random starting nodes, and we report the average performance. The table below presents these results. Performance data for all baselines, including widely used hand-crafted heuristics such as Christofides, Greedy, Nearest insertion, and Nearest-greedy, as well as the genetic-programming-based AHD method GPHH and recent LLM-based AHD approaches including EoH, ReEvo, and MCTS-AHD, are taken directly from the MCTS-AHD appendix for consistency. The heuristic produced by our framework shows a clear advantage across the benchmark suite. It attains the lowest average optimality gap among all compared methods, achieves the best tour cost on 6 instances, which is the highest win count, and outperforms the strongest heuristic found by MCTS-AHD on 9 of the 15 instances.
>
>
> **Performance of the best heuristic discovered by CALM on TSPLib instances, with the best result for each instance shown in bold**
> | Instance    | Christofides     | Greedy        | Nearest insertion | Nearest-greedy | GPHH-best | EoH     | ReEvo   | MCTS-AHD         | CALM  |
> |-------------|------------------|---------------|-------------------|----------------|-----------|---------|---------|------------------|------------------|
> | ts225       | 5.67%           | **5.38%** | 19.93%           | 16.82%        | 7.71%    | 5.57%  | 6.56%  | 10.84%          | 8.02%           |
> | rat99       | **9.43%**  | 22.30%       | 21.05%           | 21.79%        | 14.09%   | 18.78% | 12.41% | 10.46%          | 12.16%          |
> | bier127     | 13.03%          | 19.50%       | 23.05%           | 23.25%        | 15.64%   | 14.05% | 10.79% | **7.56%**  | 10.55%          |
> | lin318      | 13.80%          | 18.75%       | 24.44%           | 25.78%        | 14.30%   | 14.03% | 16.63% | 14.07%          | **13.47%** |
> | eil51       | 15.18%          | 13.03%       | 16.14%           | 31.96%        | 10.20%   | 8.37%  | 6.47%  | 15.98%          | **3.54%**  |
> | d493        | **9.52%**  | 16.68%       | 20.39%           | 24.00%        | 15.58%   | 12.41% | 13.43% | 11.73%          | 10.71%          |
> | kroB100     | 9.82%           | 16.59%       | 21.53%           | 26.26%        | 14.06%   | 13.46% | 12.20% | 11.43%          | **9.44%**  |
> | kroC100     | 9.08%           | 12.94%       | 24.25%           | 25.76%        | 16.22%   | 16.85% | 15.88% | 8.27%           | **3.87%**  |
> | ch130       | 10.09%          | 28.40%       | 19.21%           | 25.66%        | 14.77%   | 12.26% | 9.40%  | 10.18%          | **6.97**  |
> | pr299       | **11.23%** | 31.42%       | 25.05%           | 31.42%        | 18.24%   | 23.58% | 20.63% | **11.23%** | 11.73%          |
> | fl417       | 15.57%          | 12.64%       | 25.52%           | 32.42%        | 22.72%   | 20.47% | 19.15% | **10.20%** | 12.05%          |
> | kroA150     | 13.44%          | 20.24%       | 19.09%           | 26.08%        | 15.59%   | 18.36% | 11.62% | 10.08%          | **8.72%**  |
> | pr264       | **11.28%** | 11.89%       | 34.28%           | 17.87%        | 23.96%   | 18.03% | 16.78% | 12.27%          | 12.17%          |
> | pr226       | 14.17%          | 21.44%       | 28.02%           | 24.65%        | 15.51%   | 19.90% | 18.02% | **7.15%**  | 15.14%          |
> | pr439       | **11.16%** | 20.08%       | 24.67%           | 27.36%        | 21.36%   | 21.96% | 19.25% | 15.12%          | 12.81%          |
> | Average Gap | 11.50%          | 18.09%       | 23.11%           | 25.41%        | 16.00%   | 15.87% | 13.95% | 11.10%          | **10.09%** |

---

> > ### Author Response · Authors · 2025-11-20
> > **Response (2/3)**
> >
> > ## Response to Weakness 2
> >
> > Thank you for the thoughtful comment. We would first like to clarify the comparative performance in the non-GRPO setting under the same GPT-4o-mini model. Across the four evaluated tasks, CALM already shows clear advantages over MCTS-AHD on two of them without any fine-tuning. As reported in Table 1 for the OBP task, CALM achieves a lower optimality gap on 5 of the 6 test sets, and its average gap across all instances is smaller. Table 3 further shows that on the OP task, CALM consistently performs better at every problem scale.
> >
> > To address your concern, we further evaluated MCTS-AHD under Qwen with and without GRPO on both CVRP and TSP, and also assessed CALM under Qwen without GRPO, using exactly the same number of sampled responses for MCTS-AHD with GRPO (G = 4 per prompt) and the same fine-tuning hyperparameters as in our main GRPO setup so that training conditions are matched. These results lead to two key observations: (i) GRPO provides clear benefits to MCTS-AHD, bringing improvement to both tasks. On CVRP, the improvement is particularly notable, where MCTS-AHD+Qwen without GRPO performs worse than MCTS-AHD+GPT-4o-mini, but with GRPO it surpasses that baseline. (ii) CALM benefits even more strongly from GRPO than MCTS-AHD. On CVRP, CALM+Qwen without GRPO underperforms MCTS-AHD in two of three scales, yet with GRPO it outperforms MCTS-AHD across all scales. On TSP, the gain from GRPO is more moderate and does not allow MCTS-AHD+Qwen to exceed MCTS-AHD+GPT-4o-mini, likely because the seed heuristic provided in TSP is strong and leaves much less room for exploration. In contrast, applying GRPO to CALM yields a significant boost and makes CALM+Qwen surpass CALM+GPT-4o-mini. This consistent and substantial improvement suggests the effectiveness of CALM’s specialized design, such as its fine-grained operators, for RL-based fine-tuning.
> >
> > In summary, CALM is intended to work with RL-based fine-tuning methods, and our additional experiments provide further empirical indications that it tends to benefit more from GRPO, consistent with the usefulness of its specialized design choices. Without GRPO, CALM remains comparable to MCTS-AHD, which highlights its efficiency in the base setting as well.
> >
> > **More results on the CVRP task**
> >
> > | Method                    | N=50            | N=100            | N=200            |
> > |---------------------------|-----------------|------------------|------------------|
> > | MCTS-AHD (GPT-4o-mini)    | 9.372 (5.44%)  | 15.974 (6.98%)  | 28.434 (4.70%)  |
> > | CALM (GPT-4o-mini)        | 9.404 (5.81%)  | 16.046 (7.46%)  | 28.713 (5.72%)  |
> > | MCTS-AHD (Qwen, w/o GRPO) | 9.921 (11.62%) | 16.926 (13.35%) | 29.967 (10.34%) |
> > | MCTS-AHD (Qwen, w/ GRPO)  | 9.267 (4.27%)  | 15.796 (5.79%)  | 28.346 (4.37%)  |
> > | CALM (Qwen, w/o GRPO)     | 9.806 (10.32%) | 16.995 (13.82%) | 30.631 (12.78%) |
> > | CALM (Qwen, w/ GRPO)      | 9.228 (3.83%)  | 15.745 (5.44%)  | 28.23 (3.94%)   |
> >
> > **More results on the TSP task**
> > | Method                    | N=50            | N=100           | N=200            |
> > |---------------------------|-----------------|-----------------|------------------|
> > | MCTS-AHD (GPT-4o-mini)    | 6.225 (9.69%)  | 8.684 (11.79%) | 12.12 (13.71%)  |
> > | CALM (GPT-4o-mini)        | 6.273 (10.54%) | 8.691 (11.88%) | 12.104 (13.56%) |
> > | MCTS-AHD (Qwen, w/o GRPO) | 6.257 (10.26%) | 8.746 (12.58%) | 12.268 (15.09%) |
> > | MCTS-AHD (Qwen, w/ GRPO)  | 6.254 (10.19%) | 8.722 (12.28%) | 12.228 (14.72%) |
> > | CALM (Qwen, w/o GRPO)     | 6.262 (10.35%) | 8.720 (12.26%) | 12.219 (14.63%) |
> > | CALM (Qwen, w/ GRPO)      | 6.244 (10.04%) | 8.668 (11.58%) | 12.088 (13.41%) |

---

> > > ### Author Response · Authors · 2025-11-20
> > > **Response (3/3)**
> > >
> > > ## Response to Weakness 3
> > >
> > > As discussed in Section 2, a key distinction between our work and concurrent studies such as EvoTune is that CALM incorporates dedicated design choices intended to make fine-tuning more effective for AHD. In particular, CALM employs finer-grained operators, including injection and modification, to enable more targeted heuristic mutations and help RL-based methods assign credit more precisely during training. The results in the main manuscript already show that CALM outperforms EvoTune under comparable settings. In addition, the further experiments included in our rebuttal (see our response to Weakness 2) indicate that directly applying GRPO to a strong existing method like MCTS-AHD, which is not designed with fine-tuning in mind, does not yield improvements of the same scale. Overall, these observations provide empirical support that the dedicated design choices in CALM contribute to its effectiveness in the fine-tuning setting.
> > >
> > > ## Response to Question 1
> > >
> > > (1) Why hard-coded?
> > >
> > > We appreciate the reviewer’s question, which touches on a core design choice in current automatic heuristic design systems. The use of hard-coded operators reflects the mainstream paradigm adopted by nearly all existing LLM-based AHD methods. This practice originates from the tradition of genetic algorithms, where mutation and crossover operators guide the evolution of a population. In the LLM setting, these operators take the form of prompt templates that steer the search in distinct directions. Empirically, the number and diversity of such operators matter: methods that employ only a very small set of templates, such as OpenEvolve or EvoTune, generally show weaker performance across most tasks in Tables 1–3. For this reason, we follow the established and effective practice of defining an operator set.
> > >
> > > (2) Why sampling?
> > >
> > > We found that round-robin strategies used in approaches like EoH, ReEvo, or MCTS-AHD can be rigid in practice. In developing CALM, we observed that certain operators are essential for long-term improvement but do not produce immediate gains with high probability. A typical example is the simplification operator, which plays a crucial role in keeping heuristics concise and maintainable over many generations, even though it may not directly boost validation performance on the step in which it is applied. Randomized sampling allows us to assign lower but non-negligible probabilities to such operators, ensuring that they are invoked often enough to sustain long-term evolution without dominating the search.
> > >
> > > (3) Letting the LLM generate modifications.
> > >
> > > We fully agree that enabling the LLM to generate its own modifications represents an exciting direction. There is already pioneering work, such as [1], which lets the model search the AHD framework itself at a meta level. Even in that line of research, however, the system still depends on predefined prompt templates to structure the search and ensure stability. At present, all existing paradigms continue to rely on some degree of predefined operator design, and CALM follows this shared foundation. We view operator generation as a promising avenue for future work, and we appreciate the reviewer’s suggestion in this direction.
> > >
> > > [1] Qiu Z, Chen X, Chen L, et al. Mela: A metacognitive llm-driven architecture for automatic heuristic design[J]. arXiv preprint arXiv:2507.20541, 2025.
> > >
> > >
> > > ## Response to Question 2
> > >
> > > Thank you for the thoughtful question. As we discussed in Section 5.2, the slower start is plausibly linked to the more limited capability of the foundational model before any task-specific adaptation. Once training progresses, the benefits of fine-tuning begin to work, allowing CALM to overcome its initial lag.
> > >
> > > This interpretation is supported by the trends presented in Appendix I.2. The average score of the foundational LLM increases noticeably after about 100 training steps, which correspond to 400 LLM queries, and the fine-tuned model obtains its highest scores around 300 to 400 steps, which correspond to 1200 to 1600 queries. These observations indicate that the underlying heuristic generation ability of the model continues to strengthen throughout training. Consequently, although CALM may appear weaker at the earliest stage, the improving capacity of the model and the effect of fine-tuning explain why CALM ultimately surpasses other methods as training deepens.

---

> > > > ### Comment · Reviewer_9Hpb · 2025-11-25
> > > >
> > > > I thank the author for their responses, which strengthen the paper's position. I will keep my acceptance score.

---

### Official Review · Reviewer_STCg · 2025-11-01

**Soundness:** 1
**Presentation:** 3
**Contribution:** 3
**Rating:** 4
**Confidence:** 4

**Summary:**

This paper presents CALM (Co-evolution of Algorithms and Language Model), a framework for automatic heuristic design where large language models (LLMs) evolve jointly with the heuristics they generate. CALM combines evolutionary prompt search with reinforcement learning fine-tuning (GRPO) to adapt both the prompt and the model, introducing a probabilistic collapse mechanism and a relative reward function to guide stable co-evolution. Experiments on four combinatorial optimization tasks show that CALM outperforms state-of-the-art LLM-based baselines while running efficiently.

**Strengths:**

1. The related work section is clearly written and situates the method well within the context of prior research.

2. The paper provides a clear mathematical definition of the task and notations, which are often missing or ambiguous in previous work.

3. The interpretation of LLM-based evolutionary heuristic search as a general reinforcement learning process is insightful and conceptually interesting.

**Weaknesses:**

1. The use of unique token overlap in the “idea” text as a diversity metric feels superficial. Simply counting different words doesn’t necessarily indicate whether two heuristics are algorithmically or behaviorally distinct, so the diversity might be overestimated. It’s efficient, but I think something like embedding-based similarity or even code-level structure comparison would better capture the real heuristic diversity.

2. My biggest concern is still about the RL-based LLM fine-tuning. RL is known to be very sample-inefficient, and here each update needs running and scoring several heuristics, which is computationally expensive. This makes the approach hard to scale to larger problems or longer runs. Also, the performance improvement from fine-tuning (Table 4; Appendix I.2) looks quite moderate and might mainly come from other parts of the design. The fine-tuning seems to make the model more stable and consistent, but I’m not convinced it’s the main reason for the performance boost.

3. Another issue is about the experiments that mix results from GPT and Qwen models. The actual fine-tuning is only done on the open-source Qwen2.5 model with Unsloth. Since only one baseline result is available for Qwen2.5, comparing it with the GPT-4o-mini results feels a bit unfair. I understand the limitation of using closed models, but it would be better to include more baselines tested under the same Qwen model for a fairer comparison.

**Questions:**

1. The co-evolution of prompts and the LLM itself raises legitimate stability concerns, as simultaneous adaptation of two coupled components can yield non-stationary learning dynamics. CALM mitigates this through sequential update scheduling, conservative GRPO fine-tuning with KL regularization, limited LoRA updates, and a collapse mechanism that periodically resets the population (Sec. 4; Appendix C–G). These choices are reasonable and appear to prevent divergence—the training curves in Fig. 2 are smooth, and all tasks are completed successfully. However, the paper provides no explicit stability analysis or ablation contrasting co-evolution with prompt-only training, leaving the robustness of this dual-optimization setup empirically supported but theoretically uncharacterized.

---

> ### Author Response · Authors · 2025-11-20
> **Response (1/4)**
>
> ## Response to Weakness 1
>
> (1) We appreciate the reviewer’s suggestion to consider embedding‐based similarity or code‐level structural comparison, and we would like to clarify why these approaches are not well-suited in our setting. While embedding models or structural code comparison might appear more semantically meaningful than token‐level overlap, they can behave unreliably for heuristic–generation tasks where small textual edits lead to disproportionately large behavioral shifts. For instance, modifying a single symbol such as changing "$<$" to "$>$" can fundamentally alter the logic of a heuristic and produce dramatically different performance, even though the edit distance is minimal. Recent work [1] on LLM-based search has reported that embedding models struggle in such domains: although embeddings were used to project candidate solutions into a latent space for Bayesian optimization, the method consistently failed for code‐search tasks, with empirical evidence indicating that current embeddings do not provide "distinct representations for sequences with low-edit distances". These limitations make embedding-based similarity an unreliable proxy for diversity measurement in our problem.
>
> (2) In addition, we point out that using embedding models or LLMs for diversity estimation introduces practical constraints. Employing a local embedding model would require additional GPU memory, which directly conflicts with the resource demands of our reinforcement learning updates. Even fine-tuning a 7B model on a single 24GB GPU already leaves very limited memory headroom. Hosting a separate embedding model under this constraint can lead to RL update failures, especially in later stages of the search when the model generates longer responses. Using an external embedding API would introduce additional monetary cost and, importantly, may not be desirable in scenarios where the heuristic-design task involves private information and users require a fully local workflow.
>
> Overall, given both the empirical evidence that code-level embeddings fail to provide reliable distinctions and the practical constraints, we opt for a simple and lightweight idea-level metric to quantify heuristic diversity. Despite its simplicity, the results in Table 4 suggest its effectiveness.
>
> [1] Agarwal D, Arivazhagan M G, Das R, et al. Searching for optimal solutions with LLMs via bayesian optimization[C]//The Thirteenth International Conference on Learning Representations. 2025.

---

> > ### Author Response · Authors · 2025-11-20
> > **Response (2/4)**
> >
> > ## Response to Weakness 2
> >
> > (1) We would first like to clarify the computational cost of running and scoring heuristics. We agree with the reviewer that these evaluations can be expensive in some tasks. However, this cost is inherent to the broader paradigm of LLM-based automatic heuristic design that relies on heuristic scores to guide evolutionary search. Existing methods that do not fine-tune the LLM still incur the same evaluation overhead.
> >
> > (2) We also want to highlight that heuristic evaluation is not the dominant computational bottleneck. As shown in the wall-clock breakdown in Appendix I.1, LLM inference time accounts for over 70% of runtime across all tasks, while heuristic evaluation contributes a significantly smaller portion. This suggests that future scalability challenges are more likely to arise from LLM inference rather than the evaluation process itself, and that the added cost of RL-based fine-tuning does not meaningfully alter the main bottleneck.
> >
> > (3) We respectfully disagree that the benefit of RL fine-tuning is moderate. Table 4 shows that adding GRPO to CALM improves its performance (optimality gap of the discovered heuristics) at the OBP task from 1.78%, which is worse than almost all baselines, to 0.71%, achieving SOTA results. Furthermore, Appendix I.2 demonstrates that fine-tuning substantially increases feasibility rates and raises the average score of test prompts by more than 50% on both CVRP and OP tasks. These improvements indicate that fine-tuning contributes materially to the overall effectiveness of the system rather than merely stabilizing it.
> >
> > (4) We have provided more empirical evidence to show that with GRPO-based fine-tuning, both the performance of CALM and MCTS-AHD can be improved. Please refer to our response to Weakness 3 and Question 1 that you raised for details.
> >
> > (5) We acknowledge that fine-tuning may not continuously improve the LLM after many training or search steps. Importantly, however, the fine-tuning process could introduce useful variation into the heuristic generator itself. This diversity at the generator level can benefit exploration even when the absolute capability of the LLM does not monotonically increase.
> >
> > (6) Finally, even in the conservative scenario where RL-based fine-tuning primarily stabilizes the LLM’s output distribution without necessarily enabling the discovery of fundamentally new heuristics, it can still materially benefit the AHD process. By reducing variance and improving reliability, fine-tuning allows the search framework to obtain the high-quality heuristics that would eventually be found without RL but to reach them much earlier in the search trajectory. This earlier access to strong candidates, in turn, enables deeper and more productive exploration of the heuristic space under the same computational budget, because evolutionary search can begin refining higher-quality heuristics sooner.

---

> > > ### Author Response · Authors · 2025-11-20
> > > **Response (3/4)**
> > >
> > > ## Response to Weakness 3
> > >
> > > (1) We would like to clarify that our comparisons do not place our method at an unfair advantage. We sincerely appreciate the reviewer’s concern regarding mixing results from GPT and Qwen models, and we emphasize that the model we employ (Qwen2.5-7B-Instruct-INT4) is substantially weaker than GPT-4o-mini according to the Qwen2.5 technical report, which reports the approximate ordering GPT-4o-mini $\approx$ Qwen2.5-Turbo $>$ Qwen2.5-14B-Instruct $>$ Qwen2.5-7B-Instruct $>$ Qwen2.5-7B-Instruct-INT4. This means that our approach is evaluated under a less capable model, ensuring that improvements cannot be attributed to a stronger underlying LLM.
> > >
> > > (2) We conducted additional experiments under the reviewer’s thoughtful suggestion to provide fairer comparisons within the same Qwen model family. Specifically, we further evaluated MCTS-AHD under Qwen with and without GRPO on both CVRP and TSP, and also assessed CALM under Qwen without GRPO, using exactly the same number of sampled responses for MCTS-AHD with GRPO (G = 4 per prompt), the same reward function, and the same fine-tuning hyperparameters as in our main GRPO setup, so that training conditions are matched. These results lead to two key observations: (i) GRPO provides clear benefits to MCTS-AHD, bringing improvement to both tasks. On CVRP, the improvement is particularly notable, where MCTS-AHD+Qwen without GRPO performs worse than MCTS-AHD+GPT-4o-mini, but with GRPO it surpasses that baseline. (ii) CALM benefits even more strongly from GRPO than MCTS-AHD. On CVRP, CALM+Qwen without GRPO underperforms MCTS-AHD in two of three scales, yet with GRPO it outperforms MCTS-AHD across all scales. On TSP, the gain from GRPO is more moderate and does not allow MCTS-AHD+Qwen to exceed MCTS-AHD+GPT-4o-mini, likely because the seed heuristic provided in TSP is strong and leaves much less room for exploration. In contrast, applying GRPO to CALM yields a significant boost and makes CALM+Qwen surpass CALM+GPT-4o-mini. This consistent and substantial improvement suggests the effectiveness of CALM’s specialized design, such as its fine-grained operators, for RL-based fine-tuning.
> > >
> > > **More results on the CVRP task**
> > >
> > > | Method                    | N=50            | N=100            | N=200            |
> > > |---------------------------|-----------------|------------------|------------------|
> > > | MCTS-AHD (GPT-4o-mini)    | 9.372 (5.44%)  | 15.974 (6.98%)  | 28.434 (4.70%)  |
> > > | CALM (GPT-4o-mini)        | 9.404 (5.81%)  | 16.046 (7.46%)  | 28.713 (5.72%)  |
> > > | MCTS-AHD (Qwen, w/o GRPO) | 9.921 (11.62%) | 16.926 (13.35%) | 29.967 (10.34%) |
> > > | MCTS-AHD (Qwen, w/ GRPO)  | 9.267 (4.27%)  | 15.796 (5.79%)  | 28.346 (4.37%)  |
> > > | CALM (Qwen, w/o GRPO)     | 9.806 (10.32%) | 16.995 (13.82%) | 30.631 (12.78%) |
> > > | CALM (Qwen, w/ GRPO)      | 9.228 (3.83%)  | 15.745 (5.44%)  | 28.23 (3.94%)   |
> > >
> > > **More results on the TSP task**
> > > | Method                    | N=50            | N=100           | N=200            |
> > > |---------------------------|-----------------|-----------------|------------------|
> > > | MCTS-AHD (GPT-4o-mini)    | 6.225 (9.69%)  | 8.684 (11.79%) | 12.12 (13.71%)  |
> > > | CALM (GPT-4o-mini)        | 6.273 (10.54%) | 8.691 (11.88%) | 12.104 (13.56%) |
> > > | MCTS-AHD (Qwen, w/o GRPO) | 6.257 (10.26%) | 8.746 (12.58%) | 12.268 (15.09%) |
> > > | MCTS-AHD (Qwen, w/ GRPO)  | 6.254 (10.19%) | 8.722 (12.28%) | 12.228 (14.72%) |
> > > | CALM (Qwen, w/o GRPO)     | 6.262 (10.35%) | 8.720 (12.26%) | 12.219 (14.63%) |
> > > | CALM (Qwen, w/ GRPO)      | 6.244 (10.04%) | 8.668 (11.58%) | 12.088 (13.41%) |

---

> > > > ### Author Response · Authors · 2025-11-20
> > > > **Response (4/4)**
> > > >
> > > > ## Response to Question 1
> > > >
> > > > Thanks for your valuable comment. We must admit that it is difficult to provide a formal stability analysis for a system in which multiple learning components adapt together, since the interactions between prompt evolution and model updates make the overall dynamics inherently complex. Nonetheless, the paper does include empirical evidence addressing the reviewer’s concern. In the original manuscript, Table 4 already reports ablation results in which prompts evolve while the model remains fixed, corresponding to the method named "local, w/o GRPO". To strengthen this comparison, we additionally ran the OP task using Qwen without GRPO, where the results are listed in the Table below. These new experiments show that CALM combined with Qwen and GRPO achieves higher performance, reduced variance, and a statistically significant improvement according to the p-value against the GRPO-free variant. Together, these findings further support the robustness and practical stability of the dual-optimization framework.
> > > >
> > > > **Single-tailed t-tests on the OP task**
> > > > |                | run1   | run2   | run3   | run4   | run5   | run6   | run7   | run8   | run9   | run10  | avg    | std   | p-value                         |
> > > > |-----------------|--------|--------|--------|--------|--------|--------|--------|--------|--------|--------|--------|-------|---------------------------------|
> > > > | MCTS-AHD        | 14.668 | 14.910  | 14.786 | 14.602 | 14.738 | 14.724 | 14.642 | 14.826 | 14.722 | 14.690  | 14.731 | 0.091 |                                |
> > > > | CALM (w/ GRPO)  | 14.876 | 14.822 | 14.951 | 14.798 | 14.844 | 14.669 | 14.850  | 14.878 | 14.880  | 14.746 | 14.831 | 0.079 | 0.008317865 (v.s. MCTS-AHD)     |
> > > > | CALM (w/o GRPO) | 14.766 | 14.425 | 14.722 | 14.578 | 14.770  | 14.806 | 14.608 | 14.800   | 14.674 | 14.581 | 14.673 | 0.124 | 0.001545683 (v.s. CALM w/ GRPO) |

---

### Meta-Review · Area_Chair_7shQ · 2025-12-31

**Summary:**

The paper proposes CALM, an RL-augmented, LLM-based automatic heuristic design (AHD) framework that jointly evolves prompts and model parameters for improved combinatorial optimization. CALM introduces fine-grained evolutionary operators, reward shaping, a collapse mechanism, and progressive LLM fine-tuning (GRPO) to drive both diversity and performance of synthesized heuristics. Experiments on four optimization tasks demonstrate competitive improvements over recent LLM-AHD baselines.

Although the current experimental results are limited to **a finite-resource setup and INT4 quantization**, so conclusions are **significantly not mature and complete**. While the AHD field as a whole remains less developed (still focusing on prompt engineering mostly) compared to the rapidly advancing agentic/LLM research, this work offers a substantial step forward and brings new tools and reproducible baselines for the community. So I decide to accept this paper.

**Reviewer Concerns:**

Overall, most major issues were addressed, or actions were taken to mitigate reviewer concerns within a single revision round, with experiments and expanded ablations.

**Reviewer Scores:**

Reviewer ny5L has accepted to increase his score from 2 to 4. Reviewer STCg raises concerns about diversity metrics, sample efficiency, and LLM comparison fairness. Authors’ rebuttal and new results address many points. For both of them, the score (4) reflects a marginal/borderline but non-blocking stance.

---

### Decision · Program_Chairs · 2026-01-26

Accept (Poster)